# Adversarial Encoding Perturbation and Synthesis for Set Representation Auxiliary Learning

**Yankai Chen**[1,2]**, Xinni Zhang**[3]**, Henry Peng Zou**[4]**, Bowei He**[1,2]**, Yangning Li**[5]**,**
**Philip S. Yu**[4]**, Irwin King**[3]**, Xue Liu**[1,2]
[1]MBZUAI, [2]McGill University, [3]The Chinese University of Hong Kong,
[4]University of Illinois Chicago, [5]Tsinghua University

## Abstract

Sets are fundamental data structures, and learning their vectorized representations is crucial for many computational problems. While effective at preserving basic *intra-set* semantics, e.g., permutation invariance, existing methods may be insufficient in explicitly modeling *inter-set correlations*, which are critical for tasks requiring fine-grained comparisons between sets. We propose SRAL, a Set Representation Auxiliary Learning framework for capturing inter-set correlations that is compatible with various downstream tasks. SRAL conceptualizes sets as high-dimensional distributions and leverages the 2-Sliced-Wasserstein distance to derive their distributional discrepancies into set representation encoding. More importantly, we introduce a novel adversarial auxiliary learning scheme. Instead of manipulating the input data, e.g., element dropout/addition, our method introduces adversarial perturbations for encoding synthesis at the feature level. Through min-max optimization, we compel the model to achieve robustness against worst-case perturbations. Our theoretical analysis shows that this objective, in expectation, directly optimizes for the set-wise Wasserstein distances, forcing the model to learn highly discriminative representations. Comprehensive evaluations across four downstream tasks examine SRAL's performance relative to baseline methods, showing consistent effectiveness in both inter-set relation-sensitive retrieval and intra-set information-oriented processing tasks.

## 1 Introduction

Set-structured data are prevalent in practice, as they represent complex data objects composed of simpler units (Vargas-Calderón, 2025). With the rapid development of machine learning techniques, learning vectorized representations for sets is crucial (NaderiAlizadeh and Singh, 2025). It not only benefits a variety of emerging applications, e.g., similar group matching in social networks (Shen et al., 2012; Tang and Liu, 2022; Chen et al., 2023b) and object retrieval in vector databases (Lee et al., 2019; Wang et al., 2022), but also demonstrates learning-based potential in addressing data management challenges (Hadjieleftheriou et al., 2008; Wang and He, 2019; Dong and Rekatsinas, 2018; Ge et al., 2019; Li et al., 2021; Zeakis et al., 2022; Yu et al., 2025).

Traditional algorithms for retrieving set-structured data are typically *rule-based*. For example, Set Similarity Join (Arasu et al., 2006; Mann et al., 2016) computes pairs of sets from data partitions to filter candidates with a given score threshold. Methods in this area design iterative filter-verification frameworks (Deng et al., 2017; Zeakis et al., 2022) or propose index-based solutions (Li et al., 2021) to improve efficiency. However, these algorithm algorithms often lack *predictive capability* to generalize from data to make inferences. To address this limitation, *Set Representation Learning* has emerged. Unlike naive approaches that merely sum element features, its fundamental goal is to learn a holistic and fixed-size embedding that captures the intrinsic semantics of the whole unordered collection. This capability is essential for facilitating complex downstream tasks, for example, in E-commerce, enabling the model to interpret a product bundle as a cohesive semantic unit, e.g., a "camping kit", rather than a loose collection of items, thereby improving recommendation accuracy. While early machine learning approaches leveraged kernel methods (Jebara et al., 2004; Gretton et al.,

Figure 1: SRAL captures inter-set correlations for adversarial optimization (left); normalized ratios of second-best methods over SRAL are reported due to varying metric scales (right).

2006; Boiman et al., 2008; Muandet et al., 2012), recent deep learning methods focus on capturing diverse set cardinalities and ensuring invariance to element permutations (Xu et al., 2025; Skianis et al., 2020; Zaheer et al., 2017; Lee et al., 2019; Kim, 2022; Wang et al., 2023). They design a composition of permutation-equivariant neural network backbones and aggregation mechanisms to preserve set semantics that outperform traditional feature pooling techniques (Mialon et al., 2021; Murphy et al., 2018; Zhang et al., 2020). Detailed discussions are reported in Appendix B.

Despite these advancements, existing methods predominantly focus on fulfilling *intra-set properties*. While permutation invariance and cardinality independence are essential, these models are often insufficient for *explicitly capturing the rich and complex inter-set correlations*. This however is crucial for comparing set-wise similarities and differences in certain scenarios. For instance, in set retrieval tasks, identifying the nearest neighbors for a query set inherently relies on a nuanced understanding of set-to-set relationships (Naderializadeh et al., 2021; Zhang et al., 2020). Similarly, in E-commerce applications, product bundles with overlapping items appeal to similar customer segments, where capturing the subtle relationships between different product sets can potentially enhance recommendation accuracy (Ma et al., 2022). Such set-wise correlation knowledge may not be naturally inherited through the mere encoding of intra-set properties, thus creating a notable gap in representation capability.

To bridge this gap, we introduce SRAL, a **S**et **R**epresentation **A**uxiliary **L**earning framework designed to learn representations with a focus to capture inter-set correlations. Our framework formulates model optimization as a flexible auxiliary objective built upon two synergistic components. ❶ First, we introduce a novel set encoder grounded in optimal transport theory. By conceptualizing sets as empirical distributions, this encoder measures the set-wise distributional discrepancy using the 2-Sliced-Wasserstein metric (Lahn et al., 2025; Rabin et al., 2011; Bonneel et al., 2015; Villani, 2009) and derives such distance information into the set embeddings. ❷ Second, and more critically, we propose an effective adversarial auxiliary learning scheme to forge discriminative representations by training the model to resist worst-case encoding perturbations. Specifically, our approach departs from conventional data manipulation strategies, e.g., element dropout/addition or subset sampling, by introducing adversarial perturbations directly to the set features. Our theoretical analysis demonstrates that the learning over such perturbation is, in expectation, equivalent to optimizing for the 2-Sliced-Wasserstein distances between the underlying perturbed distributions. Then by training the model to be robust against worst-case perturbations via a min-max optimization, we compel the encoder to learn high quality set representations. As illustrated in Figure 1(left), these components jointly work to consolidate the fine-grained learning of inter-set correlations within a unified auxiliary framework.

To validate SRAL, we conduct extensive experiments across four diverse downstream tasks. These tasks cover both inter-set relation-sensitive retrieval applications, i.e., *Learning to Rank Set Similarity* and *Bundle Recommendation*, and intra-set information-oriented processing ones, i.e., *Point Cloud Classification* and *Topic Set Expansion*. As visualized in Figure 1 (right), SRAL not only excels at its primary goal of capturing inter-set correlations but also performs well in processing intra-set information, which underscores the effectiveness and versatility of our framework.

## 2 PRELIMINARIES

**Problem Description.** Let $\mathcal{S} = \{S_1, S_2, \ldots, S_m\}$ be a corpus of $m$ sets. The elements in these sets are drawn from a finite universe $\mathcal{E} = \{e_1, e_2, \ldots, e_n\}$, which contains $n$ unique elements. Each element $e_j \in \mathcal{E}$ is associated with a $d$-dimensional embedding vector $z_j \in \{z_1, z_2, \ldots, z_n\}$. In practice, depending on the availability of raw features, these embeddings $z_j$ are either initialized

using pre-trained feature extractors, e.g., word embeddings, or randomly initialized to be learned from scratch. The primary objective of Set Representation Learning is to learn an encoder function that maps any set $S_i$ to a fixed-size vector embedding $v_i$, which preserves inter-set correlation information to facilitate downstream set retrieval and processing tasks. All notations are explained in Appendix C.

Take bundle recommendation as a concrete example. A set $S_i$ represents a product bundle, e.g., a "camping kit", containing items such as a tent, a sleeping bag, a kettle, etc. The learning of bundle representations enables downstream systems to interpret the bundle as a coherent semantic entity that captures the collective information of the items, rather than treating them as a loose collection of isolated products. Beyond the specific e-commerce scenario illustrated above, Set Representation Learning has found broad applications across diverse domains. These include computer vision, for aggregating multi-view images or video frames (Wang et al., 2022), bioinformatics, where proteins are modeled as sets of residues for property prediction (NaderiAlizadeh and Singh, 2025), and computational pathology, where whole-slide images are treated as sets of patches for cancer diagnosis (Ilse et al., 2018; Carbonneau et al., 2018).

**Distributional Distance Measurement.** Wasserstein distance, derived from optimal transport (OT), provides a good measure for quantifying distributional distance (Tran et al., 2025; Rabin et al., 2011; Bonneel et al., 2015; Lv et al., 2024). It measures the minimum "cost" required to transform one probability distribution into another. Formally, given a probability distribution $P$, let the random variable $X$ follow the distribution $P$, i.e., $X \sim P$, $X \in \mathbb{R}^d$. For the projection function $\theta : \mathbb{R}^d \to \mathbb{R}$, $P^\theta$ represents the push-forward of $P$ with $\theta$ in a one-dimensional space, defined as $P^\theta(Y) = P(x : \theta(x) \in Y) = P(\theta^{-1}(Y))$. The $\alpha$-Wasserstein distance between $P$ and $Q$ is defined using $L^\alpha$ transport cost (Villani, 2009):

$$D_\alpha(P, Q) = \left( \inf_{g \in \text{Plans}(P,Q)} \int \|x - g(x)\|^\alpha \, dP(x) \right)^{\frac{1}{\alpha}}, \; \alpha \geq 1, \tag{1}$$

where the infimum is taken over all transport plans between $P$ and $Q$. If a minimizer exists, denoted by $g^+$, it is the solution to the OT problem. For one-dimensional distributions, a closed-form transport solution for $g^+$ from $Q$ to $P$ exists: $g^+(x) := F_P^{-1}(F_Q(x))$, where $F_Q$ and $F_P^{-1}$ denote the cumulative distribution function (CDF) and the quantile function of $P$, respectively. To prevent numerical intractability for high-dimensional cases (Kolouri et al., 2019), the alternative metric of $\alpha$-Sliced-Wasserstein distance has been recently studied (Rabin et al., 2011; Bonneel et al., 2015; Deshpande et al., 2019):

$$SD_\alpha(P, Q) = \left( \int_{\mathbb{S}^{d-1}} \left( D_\alpha(P^\theta, Q^\theta) \right)^\alpha \, d\theta \right)^{\frac{1}{\alpha}}, \; \alpha \geq 1. \tag{2}$$

$\mathbb{S}^{d-1}$ denotes the unit $d$-dimensional hypersphere. The projection is $\theta(x) = w^\top x$ where $w \in \mathbb{S}^{d-1}$ is a unit vector in $\mathbb{R}^d$. $P^\theta$ is the push-forward of $P$ using $\theta(x)$. This metric satisfies positive-definiteness, symmetry, and triangle inequality (Kolouri et al., 2016; 2019), qualifying it for similarity/distance measurement (Yang et al., 2024; Kantorovich, 1960).

## 3 SRAL FRAMEWORK

### 3.1 OVERVIEW

Our SRAL framework is designed to capture inter-set correlations through the auxiliary learning objective, which seamlessly integrates with various downstream set-based tasks. The overall learning objective combines a scenario-specific main task loss $L_{\text{Main}}$ with our auxiliary loss $L_{\text{Aux}}$, weighted by a hyper-parameter $\lambda_1$. The complete objective function is:

$$L = L_{\text{Main}} + \lambda_1 L_{\text{Aux}} + \lambda_2 \|\Xi\|_2^2, \tag{3}$$

where $\|\Xi\|_2^2$ is an $L_2$-regularizer on all trainable parameters $\Xi$ to prevent over-fitting. The core of SRAL consists of two synergistic components. First, we introduce a novel set encoder based on the 2-Sliced-Wasserstein distance, which represents sets as empirical distributions and embeds them based on their distance to a learned reference. Second, we propose a potent adversarial auxiliary learning scheme that perturbs the set features and encoding. By training the model to be robust against worst-case perturbations, this scheme forces the encoder to learn highly discriminative representations that capture fine-grained inter-set relationships.

Figure 2: SFE module illustrations: a macro and micro views of set representation encoding.

## 3.2 2-SLICED-WASSERSTEIN SET REPRESENTATION ENCODING

Our approaches begins by treating each set $S_i$ as an empirical distribution. Specifically, a set $S_i$ is represented by a list of feature vectors of its elements, $V_i = [z_{i,k} \in \mathbb{R}^d]_{k=1}^{|S_i|}$, where $z_{i,k}$ is the embedding of element $e_{i,k} \in S_i$. We assume these feature vectors are sampled from an underlying true data distribution $P_i$, and the observed features $V_i$ define the empirical distribution $\hat{P}_i$. In this work, we consider $\hat{P}_i$ to be a valid approximation of $P_i$ and unify them as $P_i$ to simplify notation.

To learn stable and discriminative set representations, our encoder leverages the distributional distance between an input set and a learnable reference distribution $O$. This reference design, characterized by $H$ trainable embeddings $V_O = [z_h \in \mathbb{R}^d]_{h=1}^H$, as a strategy shared by several previous methods (Naderializadeh et al., 2021; Guo et al., 2021a; Mialon et al., 2021), serves as a learnable "origin" in the set embedding space.

### 3.2.1 SET FEATURE ENCODER (SFE)

**Feature Mapping via Optimal Transport.** Directly computing the Wasserstein distance between high-dimensional distributions $P_i$ and $O$ is computational intractable. Therefore, we employ the 2-Sliced-Wasserstein distance, which circumvents this issue by slicing the high-dimensional distributions into multiple one-dimensional ones. Each slice is defined by a linear projection $\theta(x) = w^\top x$, where the unit vector $w$ is uniformly sampled from the hypersphere $\mathbb{S}^{d-1}$. This projection reduces $P_i$ and $O$ to their one-dimensional counterpart $P_i^\theta$ and $O^\theta$. Recall the early introduction in §2, these sliced one-dimensional distributions are compatible with the closed-form solution. As stated in (Peyré et al., 2019; Kolouri et al., 2019; Naderializadeh et al., 2021; Deshpande et al., 2019), the optimal transport map $g^+$, from the reference slice $O^\theta$ to the input slice $P_i^\theta$ can be defined as:

$$g^+(x^\theta \mid V_i^\theta) = F_{P_i^\theta}^{-1}\big(F_{O^\theta}(x^\theta)\big) \text{ where } x^\theta \in V_O^\theta, \tag{4}$$

where $V_i^\theta = [w^\top z_{i,k}]_{k=1}^{|S_i|}$ and $V_O^\theta = [w^\top z_h]_{h=1}^H$ denote the sliced features of $P_i^\theta$ and $O^\theta$, respectively. Here the CDF of $O^\theta$ is: $F_{O^\theta}(x) = \frac{1}{H} \sum_{h=1}^H \delta(x \geq w^\top z_h)$, where $\delta$ returns 1 for zero input and 0 otherwise. We include the formal statement of Eq. (4) for readability in Appendix D.

For empirical distributions defined by samples, this theoretical solution can be intuitively interpreted as a rank-matching procedure. The term $F_{O^\theta}(x^\theta)$ essentially computes the rank percentile of $x^\theta$ within the sorted values of $V_O^\theta$, and $F_{P_i^\theta}^{-1}(\cdot)$ then finds the value with the corresponding rank percentile in $V_i^\theta$. This leads to the practical implementation of $g^+$ detailed as follows: $\forall x^\theta \in V_O^\theta$, let $\tau(x \mid V)$ be the rank of value $x$ in the ordered set $V$. The mapping procedure for $g^+$ is executed as:

$$g^+(x^\theta \mid V_i^\theta) = \arg\min_{x' \in V_i^\theta} \left( \tau(x' \mid V_i^\theta) \geq \frac{|S_i|}{H} \cdot \tau(x^\theta \mid V_O^\theta) \right). \tag{5}$$

We provide its derivation in Appendix D. The indicator function $\tau(\cdot)$ can be pre-processed using the `argsort` function for $V_i^\theta$ and the `sort` function for $V_O^\theta$. Note that to accommodate cardinality differences of set features (i.e., when $H \neq |S_i|$), we employ linear interpolation, which effectively preserves the data continuity.

**Constructing `SFE` Module.** To avoid the infinite projections as required by theory in Eq. (2), we employ a Monte Carlo approximation (Kingma et al., 2013; Metropolis et al., 1953) with $R$ random projections. Let $\Theta = [w_r]_{r=1}^R$ denote all $R$ sampled projection vectors. Consequently, our Set Feature Encoder (`SFE`) module constructs the set embedding $v_i$ by aggregating the results from all projections. For each projection $w_r$, we implement the map $g^+(w_r^\top z_h \mid V_i^{\theta_r})$ for every point $z_h$ in the reference set. Following these implementations with illustration in Figure 2, SFE concatenates

along the innermost dimension to output the set embedding $v_i$ as follows:

$$\text{SFE}(V_i, V_O \,|\, \Theta) = \text{Concat}_{r=1..R;\, h=1..H}\left(g^+(w_r^\top z_h \,|\, V_i^{\theta_r})\right). \tag{6}$$

By capturing the inter-set distributional correlations, this SFE module provides the critical encoding process that we leverage in our adversarial auxiliary learning framework.

### 3.3 Adversarial Set Encoding Perturbation and Optimization

To provide adaptability across downstream tasks, we incorporate the auxiliary learning objective $L_{\text{Aux}}$ within a self-supervised paradigm. The central idea is to produce stable representations when the constituent distribution is slightly perturbed. Rather than using simple random noise, we employ an adversarial framework that compels the model to be robust against worst-case perturbations, forcing it to learn more informative inter-set information.

#### 3.3.1 Self-perturbation on Set Feature Encoding

We begin by generating perturbed samples for each set. For the input set $S_i$ with features $V_i = [z_{i,k}]_{k=1}^{|S_i|}$, the perturbation is constructed by adding small random noise to the element embeddings, where the norm of the noise is bounded by a hyper-parameter $\pi$:

$$z'_{i,k} = z_{i,k} + \epsilon'_{i,k}, \quad \text{where } \epsilon'_{i,k} \text{ is drawn from } \|\epsilon\|_2 \le \pi. \tag{7}$$

Based on the modification of $z_{i,k}$, we generate a series of perturbed distribution features, i.e., $V'_i = [z'_{i,k} \in \mathbb{R}^d]_{k=1}^{|S_i|}$. By feeding this perturbed set of elements into SFE, we obtain a perturbed set embedding $v'_i = \text{SFE}(V'_i, V_O \,|\, \Theta)$. This would allow for a more fine-grained simulation of feature variations, thereby providing high-quality perturbed samples for the subsequent learning process.

#### 3.3.2 Adversarial Min-Max Optimization

With the perturbed embeddings, we employ a self-supervised learning paradigm for optimization. To achieve this, we firstly construct two perturbed views for each set $S_i$, yielding a pair of positive set embeddings $v'_i$ and $v''_i$. We then implement with the InfoNCE loss (van den Oord et al., 2018) as follows:

$$L_{\text{wd}} = \sum_{S_i \in \mathcal{S}} -\log \frac{\exp(-\|v'_i - v''_i\|_2/\psi)}{\sum_{S_j \in \mathcal{S}} \exp(-\|v'_i - v''_j\|_2/\psi)}, \tag{8}$$

where $\psi$ is a hyper-parameter. Typically, this loss term promotes consistency between the perturbed representations of the same set $S_i$, while maximizing the Euclidean distance between embeddings of different sets, e.g., $S_i$ and $S_j$. This however raises a question: does this learning objective, which operates on perturbed embeddings, disrupt the SFE module's fundamental capability derived from the Sliced-Wasserstein metric? Therefore, we introduce Remark 1, which demonstrates that optimizing this objective is, in expectation, equivalent to optimizing the objective directly on the underlying distributional distances, thus consolidating the learning capability of our set embedding approach.

**Remark 1.** *Let $P'_i$ and $P''_i$ denote two perturbed distributions corresponding to the input set $S_i$, yielding perturbed set embeddings $v'_i$ and $v''_i$. $\mathcal{S}$, $\psi$, $SD_2$ denote the set database, a hyper-parameter, and the 2-Sliced-Wasserstein distance. For $\forall S_i \in \mathcal{S}$, we have:*

$$\mathbb{E}\left[\frac{\exp(-\|v'_i - v''_i\|_2/\psi)}{\sum_{S_j \in \mathcal{S}} \exp(-\|v'_i - v''_j\|_2/\psi)}\right] = \frac{\exp(-\|SD_2(P'_i, P''_i)\|_2/\psi)}{\sum_{S_j \in \mathcal{S}} \exp(-\|SD_2(P'_i, P''_j)\|_2/\psi)}. \tag{9}$$

With proofs in Appendix D, Remark 1 demonstrates that the Euclidean distance between the perturbed set embeddings is positively correlated in expectation with 2-Sliced-Wasserstein distance between their underlying distributions. This suggests that: by minimizing $L_{\text{wd}}$ in the embedding space, we are implicitly optimizing for the alignment of distributional distances between sets, thereby enabling the model to capture fine-grained, distribution-based inter-set correlations.

To further enhance representation robustness, we go beyond merely resisting noise but actively seek "worst-case" perturbations that maximally disrupts the representation consistency. To this end, we elevate the self-supervised objective to an adversarial min-max problem. Specifically, we seek an adversarial perturbation increment $\sigma$; $\sigma$ is shared and applied to the perturbed features $V'_i$ and $V''_i$ that are generated earlier from Eq. (7). Our goal is to find a "worst-case" perturbation that maximizes

the loss in Eq. (8). Consequently, let $\Xi$ denote all trainable parameters, our final auxiliary learning objective, $L_{\text{Aux}} = \max_{\|\sigma\|_2 \leq \pi} L_{\text{wd}}(\Xi, \sigma)$, is to be minimized under such worst-case perturbation. This derives the min-max optimization problem as follows:

$$\min_{\Xi} \max_{\|\sigma\|_2 \leq \pi} L_{\text{wd}}(\Xi, \sigma) = \min_{\Xi} \max_{\|\sigma\|_2 \leq \pi} L_{\text{wd}}(\{v_i^\sigma\}_{S_i \in \mathcal{S}}), \quad \text{where } v_i^\sigma = \text{SFE}(V_i' + \sigma, V_O \,|\, \Theta). \quad (10)$$

However, in practice, it could be computational infeasible to exactly solve the min-max problem in Eq. (10). Therefore, we employ a first-order approximation (Goodfellow et al., 2015) to efficiently estimate the optimal perturbation. We linearize the loss function $L_{\text{wd}}$ by taking its first-order Taylor expansion around $\sigma = 0$:

$$L_{\text{wd}}(\Xi, \sigma) \approx L_{\text{wd}}(\Xi, 0) + \sigma^\top \nabla_\epsilon L_{\text{wd}}(\Xi; \epsilon)\big|_{\epsilon=0}. \quad (11)$$

Maximizing this linear approximation under the norm constraint has a closed-form solution where the perturbation $\sigma$ is aligned with the gradient direction. This motivates us to approximate the worst-case perturbation with a single step of gradient ascent and thus decompose the min-max problem with the following two alternating steps:

1. **Inner Maximization to Find $\sigma$.** With the model parameters $\Xi$ fixed, we compute the gradient of the loss with respect to a small perturbation $\epsilon$ evaluated at $\epsilon = 0$:

$$g_\sigma = \nabla_\epsilon L_{\text{wd}}(\Xi; \epsilon)\big|_{\epsilon=0}. \quad (12)$$

We then update the perturbation along the gradient direction to obtain an initial adversarial perturbation $\hat{\sigma}$ as $\hat{\sigma} = \eta \cdot g_\sigma$, where $\eta$ is the ascent step size. To satisfy the constraint, we project this perturbation back onto the $\ell_2$ ball of radius $\pi$:

$$\sigma = \hat{\sigma} \cdot \min\left(1, \frac{\pi}{\|\hat{\sigma}\|_2}\right). \quad (13)$$

2. **Outer Minimization to Update $\Xi$.** After identifying the adversarial perturbation increment $\sigma$, we apply it to generate the final perturbed embeddings, e.g., $v_i^\sigma = \text{SFE}(V' + \sigma, V_O \,|\, \Theta)$. Then, we compute the adversarial loss based on these examples, which is further combined with the main task loss $L_{\text{main}}$. With $\beta$ as the learning rate, the entire model's parameters $\Xi$ are updated via gradient descent:

$$\Xi \leftarrow \Xi - \beta \cdot \nabla_\Xi \big(L_{\text{Main}} + \lambda_1 L_{\text{adv}} + \lambda_2 \|\Xi\|_2^2\big). \quad (14)$$

Through this procedure, our method learns to acquire stable set representations from deliberate feature perturbation, with the following remark to formalize this intuition (proofs are in Appendix D).

**Remark 2.** *Our min-max optimization objective in Eq. (10) is approximately equivalent to an implicit regularization of the SFE's local Lipschitz continuity for representation stability.*

Generally, our method differentiates from conventional methods by adversarial perturbing the encoding process, and thus showcases its effectiveness and convergence efficiency in §4.3.2.

## 4 EXPERIMENTS

### 4.1 SETUPS

**Tasks and Datasets.** We conduct comprehensive experiments across four diverse downstream tasks. These tasks are selected to span two primary categories: inter-set relation-sensitive tasks, i.e., Set Similarity Ranking, Bundle Recommendation, which require a deep understanding of correlations between different sets, and intra-set information-oriented tasks, i.e., Point Cloud Classification, Topic Set Expansion, which focus on processing the internal contents of a single set. All tasks and datasets conducted in our experiments are in a supervised setting. Due to the page limit, we supplement the detailed introduction of tasks and datasets in Appendix E.

- **Task 1: Learning to Rank Set Similarity.** This task evaluates the model's ability to learn from known similar set pairs and predict new associations by ranking sets based on the Euclidean distance of their learned embeddings. We use two large-scale, real-world social network datasets: Friendster (Yang and Leskovec, 2015) and LIVEJ (Mislove et al., 2007).
- **Task 2: Bundle Representation Learning for Recommendation.** In this e-commerce scenario, the goal is to recommend bundles (sets of items) to users. Effective bundle representation is crucial for prediction accuracy. Experiments are conducted on the Youshu (Chen et al., 2019) book bundle dataset and the NetEase (Cao et al., 2017) for music playlist recommendation.

Table 1: Performance comparison for Tasks 1 (left) and 2 (right). Best and second-best cases are highlighted. Statistically significant improvements ($p < 0.05$) are marked with $*$.

**Task 1: Set Similarity Learning**

| Model | Friendster | | | | LIVEJ | | | |
|---|---|---|---|---|---|---|---|---|
| | R@20 | N@20 | R@100 | N@100 | R@20 | N@20 | R@100 | N@100 |
| SAP | 72.41 | 68.13 | 85.32 | 72.15 | 79.86 | 77.75 | 86.94 | 85.34 |
| SMP | 70.78 | 68.99 | 81.61 | 72.22 | 78.95 | 76.70 | 86.83 | 84.65 |
| DeepSet | 63.20 | 60.75 | 76.60 | 69.89 | 75.45 | 74.55 | 83.31 | 79.76 |
| RepSet | 80.63 | 76.56 | 86.49 | 74.92 | 82.15 | 79.63 | 88.41 | 83.12 |
| SAtt | 77.52 | 71.92 | 87.51 | 75.21 | 83.79 | 81.73 | 91.39 | 85.07 |
| PoT | 82.44 | 81.85 | 86.96 | 81.47 | 83.18 | 84.25 | 89.33 | 86.45 |
| Set2Box | 67.35 | 69.73 | 73.46 | 70.33 | 77.24 | 75.89 | 85.12 | 82.34 |
| OTKE | 79.53 | 73.68 | 86.64 | 79.59 | 81.45 | 79.82 | 87.95 | 85.10 |
| DIEM | 82.49 | 81.40 | 88.36 | 81.56 | 83.95 | 84.92 | 89.88 | 87.15 |
| PSWE | 83.05 | 84.26 | **88.59** | 85.77 | 83.52 | 84.61 | 89.48 | 86.67 |
| FSPool | 79.90 | 81.96 | 87.76 | 84.41 | **85.36** | **87.17** | **93.07** | **90.29** |
| FSW | **83.58** | **84.39** | 88.52 | **85.81** | 84.19 | 85.04 | 89.95 | 87.23 |
| **SRAL** | **91.57** | **92.22** | **94.53** | **93.01** | **87.56** | **89.31** | **92.93** | **91.25** |
| Gain | 9.56%$^*$ | 9.28%$^*$ | 6.71%$^*$ | 8.39%$^*$ | 2.58%$^*$ | 2.46%$^*$ | -0.15% | 1.06%$^*$ |

**Task 2: Bundle Recommendation**

| Model | Youshu | | | | NetEase | | | |
|---|---|---|---|---|---|---|---|---|
| | R@20 | N@20 | R@100 | N@100 | R@20 | N@20 | R@100 | N@100 |
| MFBPR | 19.97 | 11.67 | 44.33 | 17.95 | 5.21 | 2.98 | 14.15 | 4.92 |
| DSBRec | 20.46 | 12.03 | 45.34 | 18.12 | 5.51 | 3.04 | 14.76 | 5.14 |
| DAM | 20.83 | 11.99 | 45.58 | 18.38 | 5.54 | 3.11 | 14.98 | 5.12 |
| BundleNet | 22.85 | 11.90 | 47.84 | 19.19 | 6.17 | 3.44 | 16.26 | 5.83 |
| BGCN | 25.22 | 14.54 | 49.38 | 21.18 | 7.04 | 3.91 | 17.25 | 6.51 |
| CrossCBR | 26.41 | 16.55 | 51.90 | 23.30 | 7.21 | 4.08 | 18.32 | 6.77 |
| **SRAL+** | **26.92** | **16.95** | **52.18** | **23.64** | **7.37** | **4.21** | **18.66** | **7.01** |
| Gain | 1.93%$^*$ | 2.42%$^*$ | 0.54%$^*$ | 1.46%$^*$ | 2.22%$^*$ | 3.19%$^*$ | 1.86%$^*$ | 3.54%$^*$ |

- **Task 3: Point Cloud Processing.** A point cloud is a set of 3D data points representing an object's surface. This task aims to classify the object category based on its point cloud representation. We utilize the standard ModelNet40 (Wu et al., 2015) benchmark dataset.
- **Task 4: Topic Set Expansion.** Given a small seed set of keywords describing a topic, the objective is to expand this set with other semantically related keywords from a vocabulary. We use the LDA-1k, LDA-3k, and LDA-5k datasets (Zaheer et al., 2017) with different sizes and scopes.

**Implementation Configurations.** For all tasks, we employ a consistent procedure for data preparation and training: we partition the datasets into training and testing sets with an 8:2 ratio. The training set is then further subdivided into training and validation subsets using an 8:2 split to facilitate hyper-parameter tuning. All reported results are the average of five independent runs. Hyper-parameters and experiment configurations for reproducing are reported in Appendix E.1–E.2.

**Competing Methods.** We compare SRAL against a comprehensive list of baselines for four tasks. For the general set representation tasks, i.e., Set Similarity, Point Cloud Classification, and Topic Set Expansion, we include classic pooling methods SAP (Lin et al., 2013), SMP (Lin et al., 2013) and state-of-the-art deep learning models such as DeepSet (Zaheer et al., 2017), RepSet (Skianis et al., 2020), SAtt (Lee et al., 2019), PoT (Guo et al., 2021a), Set2Box (Lee et al., 2022), OTKE (Mialon et al., 2021), DIEM (Kim, 2022), FSPool (Zhang et al., 2020), PSWE (Naderializadeh et al., 2021), and FSW (Amir and Dym, 2025). For the specialized Bundle Recommendation, we compare against established recommendation models: MFBPR (Rendle et al., 2012), DSBRec (Zaheer et al., 2017), DAM (Chen et al., 2019), BundleNet (Deng et al., 2020), BGCN (Chang et al., 2020), and the state-of-the-art CrossCBR (Ma et al., 2022). For this task, we integrate SRAL into CrossCBR to enhance its bundle embedding module, denoting it as SRAL+. Detailed descriptions are in Appendix E.

## 4.2 EVALUATION RESULTS AND DISCUSSIONS

**Task 1: Learning to Rank Set Similarity.** We evaluate all models on Task 1, where sets are ranked based on Euclidean distance in their embedding space. The results for Recall (R@k) and NDCG (N@k) are summarized in Table 1 (left) with threefold observations. ❶ Deep set encoders, e.g., DeepSet, RepSet, SAtt, generally outperform conventional pooling methods like SAP and SMP. Moreover, general OT-based methods, e.g., PoT and OTKE, and particularly Sliced-Wasserstein-based methods such as PSWE and FSW, further elevate the overall performance. ❷ Our proposed SRAL consistently achieves state-of-the-art performance across nearly all metrics and datasets. On the Friendster dataset, SRAL improvements over the best baseline ranging from 6.71% to 9.56%. On LIVEJ, it also demonstrates advantages, particularly in capturing top-ranked items with gains of 2.58% in R@20 and 2.46% in N@20. ❸ The vast majority of SRAL's performance gains are statistically significant, confirmed by a Wilcoxon signed-rank test (Conover, 1999) at a 95% confidence level. For complete results, please refer to Appendix E.3.3.

**Task 2: Bundle Representation Learning for Recommendation.** The problem of this task is formulated with a prediction function between bundle and user embeddings. Our analyses from Table 1 (right) yield three key findings. ❶ SRAL+ consistently outperforms baseline models on both Youshu and NetEase datasets, where the improvements are statistically significant across all metrics.

Table 2: Performance comparison for Task 3: Point Cloud Processing.

| | | | | | | Task 3: Point Cloud Processing | | | | | | | |
|---|---|---|---|---|---|---|---|---|---|---|---|---|---|
| Backbone | SAP | SMP | RepSet | SAtt | PoT | Set2Box | OTKE | DIEM | PSWE | FSPool | FSW | SRAL | Gain |
| MLP | 57.65 | 86.35 | 83.45 | 85.89 | 85.20 | 82.15 | 85.92 | 85.58 | **86.41** | 85.76 | 86.38 | **86.53** | +0.14% |
| ISAB | 85.45 | 86.82 | 86.05 | 86.78 | 86.55 | 85.88 | 86.70 | 86.72 | 86.85 | 86.88 | **86.93** | **87.31** | +0.44%* |

Table 3: Performance comparison for Task 4: Topic Set Expansion.

| | | | | | | | Task 4: Topic Set Expansion | | | | | | | |
|---|---|---|---|---|---|---|---|---|---|---|---|---|---|---|
| Data | SAP | SMP | DeepSet | RepSet | SAtt | PoT | Set2Box | OTKE | DIEM | PSWE | FSPool | FSW | SRAL | Gain |
| LDA-1k | 54.34 | 67.21 | 54.98 | 57.32 | 58.55 | 58.94 | 50.59 | 62.95 | 63.58 | 58.36 | **75.67** | 64.56 | **80.94** | +6.96%* |
| LDA-3k | 51.95 | 74.40 | 51.96 | 58.33 | 77.48 | 73.40 | 64.98 | 77.59 | 75.67 | 78.44 | 70.57 | **79.67** | **87.93** | +10.37%* |
| LDA-5k | 51.34 | 80.65 | 52.05 | 61.39 | 74.59 | 75.11 | 65.67 | 72.57 | 76.96 | 78.81 | 71.16 | **80.94** | **86.20** | +6.50%* |

❷ We attribute this to the strength of CrossCBR backbone, which already excels at capturing user-bundle collaborative filtering signals (Rendle et al., 2012; He et al., 2017). Our SRAL+ enhances this by providing a complementary signal, an explicit and semantically rich representation of the internal bundle structure. ❸ We recognize that this task ultimately aims to predict the user-bundle matching probability via learning the user-bundle interactions; therefore, this suggests that a more integrated learning framework is a promising direction for future work, where both bundle semantics and collaborative patterns are co-optimized.

**Task 3: Point Cloud Processing.** We evaluate various set encoding methods on top of two distinct backbones: a Multi-layer Perceptron (MLP) and the more advanced Induced Set Attention Block (ISAB) (Lee et al., 2019). The test accuracies are reported in Table 2. ❶ We observe that SRAL achieves favorable results when paired with both backbones. With the standard MLP, SRAL yields a competitive accuracy of 86.53%. When the ISAB backbone is employed, the performance margin widens, and SRAL further reaches a statistically significant improvement of +0.44% over the second-best model FSW. ❷ While using ISAB backbone clearly improves performance for all methods, SRAL provides an additional performance lift in both settings. This indicates that SRAL is effective with less impact from the backbone model selection and compatible with different feature extractors.

**Task 4: Topic Set Expansion.** With vocabularies encoded by word2vec (Mikolov, 2013), models classify elements based on their semantic similarity to a given query set. As shown in Table 3, SRAL consistently outperforms baselines in distinguishing intra-set semantics for topic expansion with AUC improvements ranging from 6.50% to 10.37%, demonstrating its effectiveness in capturing latent set semantics for classification. More importantly, in addition to its competitive performance on inter-set relation-sensitive applications (Task 1 & 2), SRAL also achieves superior results in intra-set information-oriented ones (Task 3 & 4), demonstrating its versatility as a good auxiliary representation learner.

## 4.3 EMPIRICAL ANALYSES OF SRAL

In this section, we delve into the design choices of SRAL through a series of empirical analyses. We use the Friendster dataset from Task 1 as the primary testbed for these studies.

### 4.3.1 STUDY OF SRAL SET ENCODING DESIGN

**Comparison with Other Designs.** The design of our Set Feature Encoding (SFE) module is fundamentally based on the Wasserstein distance, which benefits from its implicit regularization to the embedding distance. To validate this design choice, we compare its performance against other common distributional distance metrics, namely the Kullback-Leibler (KL) Divergence, Jensen-Shannon (JS) Divergence, and the Sinkhorn Distance. Implementation details are reported in Appendix E.7. The results in Figure 3 (A) lead to two primary conclusions. ❶ SRAL and the Sinkhorn-based method emerge as the most competitive approaches. This highlights the overall superiority of the Wasserstein distance for this task. Furthermore, SRAL achieves the best performance, which we attribute to our novel SFE module that can capture complex set similarities more effectively than simpler methods. ❷ We observe that our method incurs a relatively higher computational cost, i.e., time cost per training epoch. However, considering the performance improvement it delivers, we view this as an acceptable trade-off between model effectiveness and computational efficiency.

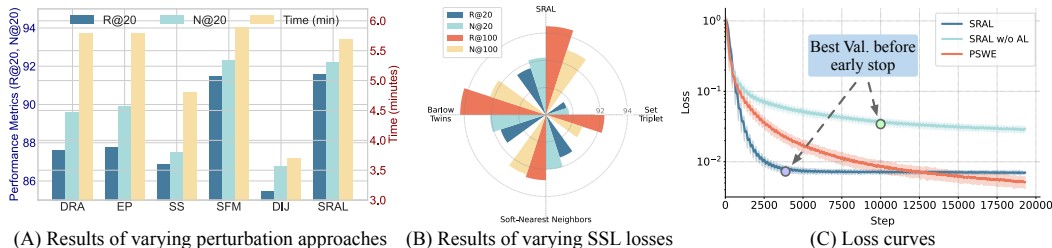

(A) Results of varying perturbation approaches    (B) Results of varying SSL losses    (C) Loss curves

Figure 4: Results of varying (A) perturbation approaches, (B) SSL losses, and (C) loss curves.

**Setting $R$ and $H$.** We investigate the impact of Monte Carlo trials $R$ and the reference feature length $H$, with the results presented in Figure 3 (B). ❶ As observed, the model's performance improves as both $R$ and $H$ increase. The performance is particularly sensitive to $R$. For instance, when fixing $H$ at 32, increasing $R$ from 4 to 32 leads to a significant rise in Recall@20 from 41.23% to 91.57%, which is attributed to a more accurate approximation of the cumulative distribution. But the performance improvement shows diminishing marginal returns at larger parameter values, especially as the performance curve begins to plateau for $R > 32$. ❷ We also notice that varying $H$

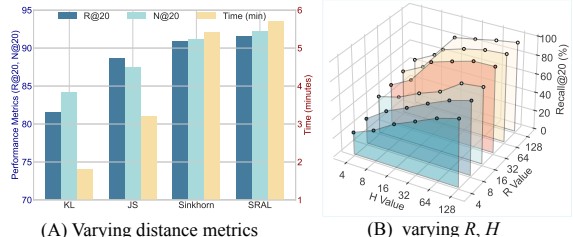

(A) Varying distance metrics     (B) varying $R$, $H$

Figure 3: Results of varying (A) distance metrics and (B) $R$ and $H$ values.

based on fixed $R$ provide less significant impact. Therefore, we select $H = 32$ and $R = 128$ as our final configuration to strike an ideal balance between model performance and resource consumption.

### 4.3.2 STUDY OF ADVERSARIAL ENCODING PERTURBATION AND OPTIMIZATION

**Perturbation Approaches.** We introduce: ❶ Element-level Perturbation as random dropout or addition of set elements (DRA), and element replacement (EP). ❷ Set-level Perturbation as subset sampling (SS) (Yun et al., 2019), and set feature mixing (SFM) (Zhang et al., 2017). ❸ Noise Injection as direct injection of noise (DIJ) into the encoded set embeddings. As shown in Figure 4 (A), both SRAL and SFM achieve competitive performance, outperforming simple data manipulation methods like EP and SS. This suggests that perturbing the set features yields more fine-grained and effective augmentations. Furthermore, compared to DIJ that directly perturbs final set embeddings, our strategy of perturbing the intermediate encoding process demonstrates superior effectiveness. Finally, while SRAL's performance is on par with SFM, it is slightly more efficient (5.7min vs. 5.9min). This is because our approach perturbs individual sets, whereas SFM requires mixing multiple sets, allowing SRAL to achieve a better balance between performance and efficiency.

**Implementation of Self-supervised Learning Loss.** While the specific self-supervised learning (SSL) objective in Eq. (8) is not the primary focus of this paper, we investigate the flexibility of our framework with other loss functions. We tested several alternatives, including Set Triplet Loss, Soft-Nearest Neighbors Loss (Frosst et al., 2019), and Barlow Twins Loss (Zbontar et al., 2021). The detailed formulations of these objectives can be found in Appendix E.7. As presented in Figure 4 (B), the results demonstrate that our framework is compatible with various SSL objective functions, as these variants achieve competitive performance.

**Convergence Analysis of Adversarial Encoding Optimization.** Figure 4 (C) presents the loss curves to illustrate model convergence. Although SRAL may have a higher complexity, compared to its ablated version without adversarial learning ("w/o AL"), we observe that SRAL converges faster and deeper in practice, reaching its best validation performance early in training. The baseline method requires a longer training duration and exhibits greater volatility. This experiment demonstrates SRAL's practical efficiency, effectively mitigating potential concerns about its computational overhead due to its rapid convergence. A more detailed analysis of model scalability is provided in Appendix E.8.

Table 4: Ablation study.

| Variant | Task 1 | | Task 2 | | Task 3 | | Task 4 |
|---|---|---|---|---|---|---|---|
| | R@20 | N@20 | R@20 | N@20 | M-ACC | I-ACC | AUC |
| w/o SFE | 67.02 (-26.81%) | 69.38 (-24.77%) | 25.31 (-5.98%) | 15.14 (-10.68%) | 66.45 (-23.20%) | 72.48 (-16.99%) | 73.39 (-16.54%) |
| w/o LI | 75.45 (-17.60%) | 74.29 (-19.44%) | 25.96 (-3.57%) | 15.83 (-6.61%) | 74.77 (-13.59%) | 79.75 (-8.66%) | 72.44 (-17.62%) |
| w/o AEPO | 77.13 (-15.77%) | 79.42 (-13.88%) | 26.22 (-2.60%) | 16.38 (-3.36%) | 86.56 (+0.03%) | 86.46 (-0.97%) | 66.21 (-24.70%) |
| w/o AL | 87.38 (-4.58%) | 88.86 (-3.64%) | 26.94 (+0.07%) | 16.97 (+0.11%) | 86.37 (-0.18%) | 87.27 (-0.04%) | 83.53 (-5.00%) |
| **SRAL** | **91.57** | **92.22** | **26.92** | **16.95** | **86.53** | **87.31** | **87.93** |

### 4.3.3 ABLATION STUDY

We evaluate several ablation variants across four tasks. Due to space constraints, we report results for Friendster, Youshu, LDA-3k on Tasks 1, 3 (R@20 and N@20) and Task 4 (AUC), while Task 2 reports accuracy with MLP and ISAB backbones (M-ACC and I-ACC). From Table 4, we observe that: ❶ replacing SFE module with mean-pooling ("w/o SFE") results in substantial performance degradation despite the auxiliary learning with our encoding perturbation. This clearly validates the effectiveness and necessity of our SFE for capturing complex set features. ❷ Substituting linear interpolation in Eq. (3) with a two-layer MLP ("w/o LI") yields less reliable dimension completion compared to our implementation. ❸ Variant "w/o AEPO" retains the auxiliary learning objective but disables the adversarial optimization step. Specifically, we utilize the inner InfoNCE loss but remove the min-max strategy that generates worst-case perturbations via gradient ascent. The results show that it negatively impacts performance, particularly on Task 4 where the AUC drops by 24.70%. ❹ In contrast, removing the entire auxiliary learning ("w/o AL") by training the model solely with the main task supervision, results in a marginal performance improvement for Task 2.

We attribute this to the specific nature of SRAL+, where its backbone model CrossCBR's graph structure is already highly optimized for capturing user-bundle collaborative signals. Our adversarial learning, which focuses on the bundle-side representation by modeling inter-bundle relationships, may provide limited complementary information for interaction prediction goal. This suggests a more integrated framework for this interaction-centric scenario.

## 5 CONCLUSION AND FUTURE WORK

We introduced SRAL, an effective auxiliary learning framework for Set Representation Learning tha is compatible with set-based retrieval and processing problems. We first introduce a set encoder based on the 2-Sliced-Wasserstein distance, which effectively captures distributional discrepancies between sets. We then propose an adversarial learning paradigm that strengthens representations by generating and optimizing worst-case perturbations to set features and encoding. Experiments demonstrate the performance superiority of SRAL over competing methods and the efficacy of its constituent design components. A promising future direction is to investigate the integration of SRAL with approximate nearest neighbor search algorithms, such as set-vector index construction algorithm and index-based search approaches (Johnson et al., 2019), for efficient online retrieval settings. Another direction is to explore the synergy with Large Language Models (LLMs), such as utilizing context-aware LLM representations as element features, or employing SRAL as a structural adapter to generate compact soft prompts for LLM-based set reasoning.

### ACKNOWLEDGMENTS

This work was supported in part by the Research Grants Council of the Hong Kong Special Administrative Region, China (CUHK 2410072, RGC R1015-23), (CUHK 2300246, RGC C1043-24G), (CUHK 7010870), and by NSF under grants III-2106758 and POSE-2346158.

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

## A  THE USE OF LARGE LANGUAGE MODELS (LLMS)

In preparing this manuscript, we employed a Large Language Model (LLM) as an assistive tool mainly for refining prose for grammar, clarity, and conciseness. The LLM did not contribute to research conception, methodological development, result analysis, or scientific conclusions. All content, including final text and code, was thoroughly reviewed, edited, and validated by the authors, who retain full responsibility for the work's accuracy and integrity.

## B  EXTENDED RELATED WORK

**Set Representation Learning.** Traditional set-based problems, e.g., set similarity join and search, have long relied on rule-based algorithms to process set data efficiently (Arasu et al., 2006; Mann et al., 2016; Li et al., 2021; Deng et al., 2017; Zeakis et al., 2022). To enable predictive modeling, early learning-based approaches often adopt kernel methods for sets or distributions (Jebara et al., 2004; Gretton et al., 2006; Boiman et al., 2008; Muandet et al., 2012). More recently, a large body of work learns permutation-invariant mappings via pooling or attention mechanisms (Zaheer et al., 2017; Murphy et al., 2018; Skianis et al., 2020; Lee et al., 2019; Zhang et al., 2019; 2020). While some models learn an optimal permutation for prediction (Zhang et al., 2019; Rezatofighi et al., 2018), others define a canonical ordering through sorting-based operators (Zhang et al., 2020). Set Transformer leverages Transformer-style attention for permutation-invariant set modeling (Lee et al., 2019; Vaswani et al., 2017), and Perceiver further designs attention-based pooling where cross-attention acts as the permutation-invariant aggregator (Jaegle et al., 2021). Beyond pooling/attention, trainable optimal-transport embeddings have also been explored for feature aggregation (Mialon et al., 2021). Other representation families include similarity-preserving box embeddings for sets (Lee et al., 2022) and measure-theoretic compact fuzzy set representations (Xu et al., 2025).

RepSet is among the earliest approaches that explicitly incorporates optimal transport / matching structure into set representation learning by solving a bipartite matching problem between input elements and learnable hidden sets (Skianis et al., 2020). However, unlike this high-dimensional exact matching formulation, our SRAL employs a 2-Sliced-Wasserstein metric to bypass the computational bottleneck via efficient approximation (Bonneel et al., 2015; Kolouri et al., 2019; Chen et al., 2024). Recent models also consider meta-learning settings (Guo et al., 2021a; Lee et al., 2023), and pooling by sliced-Wasserstein embedding has shown strong empirical performance (Naderializadeh et al., 2021). Amir and Dym propose the Fourier sliced-Wasserstein (FSW) embedding to provide injective or bi-Lipschitz representations for multisets and measures (Amir and Dym, 2025). In addition, quantum deep set modeling has been investigated (Vargas-Calderón, 2025), and optimal-transport aggregation has been applied to residue-level protein language model embeddings (NaderiAlizadeh and Singh, 2025).

**Self-supervised Learning.** Self-supervised learning generates augmented samples and derives supervision signals to guide representation learning. Contrastive frameworks encourage similar views to cluster while pushing dissimilar ones apart (Chen et al., 2020c; He et al., 2020; Chen et al., 2020a;b; Dwibedi et al., 2021; He et al., 2025; Zhang et al., 2024). Non-contrastive methods can also learn meaningful representations without feature collapse; for example, BYOL minimizes positive-pair distances with a momentum encoder (Grill et al., 2020), and Barlow Twins reduces redundancy by optimizing cross-correlation statistics between views (Zbontar et al., 2021). While most approaches manipulate input data, recent work modifies latent representations directly (Yüksel et al., 2021; Yu et al., 2022). Our work investigates perturbing the set encoding process, providing theoretical guarantees and outperforming existing augmentation designs.

**Distributional Distance Measurement.** Common distributional discrepancies include Kullback–Leibler divergence (Kullback and Leibler, 1951), Jensen–Shannon divergence (Endres and Schindelin, 2003), and Hellinger distance (Hellinger, 1909). The Wasserstein metric (Kantorovich, 1960; Villani, 2009), due to its strong theoretical properties, has gained increasing attention in modern machine learning (Arjovsky et al., 2017; Tolstikhin et al., 2018). However, exact Wasserstein computation is often intractable in high dimensions. Optimization-based approximations exist, such as entropic regularization (Sinkhorn) (Cuturi, 2013) and convolutional Wasserstein distances on geometric domains (Solomon et al., 2015). To further improve efficiency, sliced-Wasserstein distances project high-dimensional distributions onto multiple one-dimensional subspaces and exploit closed-form

solutions (Bonneel et al., 2015; Kolouri et al., 2019; Chen et al., 2023a). This line has been applied across domains, including sliced-Wasserstein kernels and flows (Kolouri et al., 2016; Liutkus et al., 2019), and also underpins our method for modeling inter-set distributional similarities (Naderializadeh et al., 2021; Peyré et al., 2019).

## C   NOTATION EXPLANATIONS

We explain the key notations in Table 1.

## D   THEORETICAL PROOFS

**Complete Formulation for Eq. (4).**   Let $V_i^\theta = [w^\top z_{i,k}]_{k=1}^{|S_i|}$ and $V_O^\theta = [w^\top z_h]_{h=1}^{H}$ denote the sliced features of $P_i^\theta$ and $O^\theta$, respectively. For any projected input $x^\theta \in V_O^\theta$, the OT solver is formulated as:

$$g^+(x^\theta \mid V_i^\theta) = F_{P_i^\theta}^{-1}\Big(F_{O^\theta}(x^\theta)\Big). \tag{1}$$

*Proof.* The transport distance, as one candidate plan for the $\alpha$-Wasserstein distance (Eq.(1) in Section 2), can be computed as

$$\text{Distance}(g^+) = \left( \int \|x^\theta - g^+(x^\theta \mid V_i^\theta)\|^\alpha \, dO^\theta(x^\theta) \right)^{1/\alpha}$$

$$= \left( \int \|x^\theta - F_{P_i^\theta}^{-1}(F_{O^\theta}(x^\theta))\|^\alpha \, dO^\theta(x^\theta) \right)^{1/\alpha}$$

$$= \left( \int_0^1 \|F_{O^\theta}^{-1}(y) - F_{P_i^\theta}^{-1}(y)\|^\alpha \, dy \right)^{1/\alpha} = D_\alpha(O^\theta, P_i^\theta), \tag{2}$$

which equals $D_\alpha(P_i^\theta, O^\theta)$ by symmetry. This proves to be the optimal distance for the two slices. $\square$

**Mapping Procedure for $g^+$.**   Let $\tau(x^\theta \mid V)$ be the rank of value $x^\theta$ in the ordered set $V$. The mapping procedure for $g^+$ is executed as follows:

$$g^+(x^\theta \mid V_i^\theta) = \begin{cases} \arg\min_{x' \in V_i^\theta} \left( \tau(x' \mid V_i^\theta) = \tau(x^\theta \mid V_O^\theta) \right), & \text{if } H = |S_i|, \\ \arg\min_{x' \in V_i^\theta} \left( \tau(x' \mid V_i^\theta) \geq \frac{|S_i|}{H} \cdot \tau(x^\theta \mid V_O^\theta) \right), & \text{if } H \neq |S_i|. \end{cases} \tag{3}$$

*Proof.* The empirical distributions of $V_i^\theta$ and $V_O^\theta$, e.g.,

$$F_{O^\theta}(x^\theta) = \frac{1}{H} \sum_{h=1}^{H} \delta(x^\theta = w^\top \cdot z_h), \tag{4}$$

are monotonically increasing. If $H = |S_i|$, we can firstly modify the original form of the optimal transport map $F_{P_i^\theta}^{-1}\big(F_{O^\theta}(x^\theta)\big)$ to:

$$g^+(x^\theta \mid V_i^\theta) = \arg\min_{x' \in V_i^\theta} \left( F_{P_i^\theta}(x') = y \right) \text{ where } y = F_{O^\theta}(x^\theta). \tag{5}$$

$\tau(x^\theta \mid V_O^\theta)$ is the ranking of each input $x^\theta$ in the ascending sorting of $V_O^\theta$, and then we can quantitatively replace the term $F_{P_i^\theta}(\cdot)$:

$$g^+(x^\theta \mid V_i^\theta) = \arg\min_{x' \in V_i^\theta} \left( \tau(x' \mid V_i^\theta) = \tau(x^\theta \mid V_O^\theta) \right) \quad \text{if } H = |S_i|. \tag{6}$$

If $H \neq |S_i|$, in this work applies the linear interpolation to the sorted positions. Therefore, $\dfrac{|S_i|}{H}$ is set as the interpolation point to complete the proof. $\square$

Table 1: Notation Table.

| Notation | Description |
|---|---|
| **Basic Sets and Elements** | |
| $\mathcal{E} = \{e_1, \cdots, e_n\}$ | All elements. |
| $\mathcal{S} = \{S_1, \cdots, S_m\}$ | A corpus of sets. |
| $S_i = \{e_{i,k} \in \mathcal{E}\}_{k=1}^{\lvert S_i \rvert}$ | Set $S_i$ containing $\lvert S_i \rvert$ elements. |
| $V_i = [z_{i,k} \in \mathbb{R}^d]_{k=1}^{\lvert S_i \rvert}$ | Features for the input set $S_i$. |
| $v_i$ | The final vector embedding for the set $S_i$. |
| **Distributions and Optimal Transport** | |
| $P_i$ | The underlying distribution of elements for set $S_i$. |
| $O$ | A trainable reference distribution. |
| $V_O = [z_h \in \mathbb{R}^d]_{h=1}^{H}$ | Features for the reference distribution $O$. |
| $P_g$ | The push-forward distribution of $P$ by function $g$. |
| $D_\alpha(P, Q)$ | The $\alpha$-Wasserstein distance between distributions $P$ and $Q$. |
| $SD_\alpha(P, Q)$ | The $\alpha$-Sliced-Wasserstein Distance between $P$ and $Q$. |
| $g^+$ | The optimal transport plan. |
| $\mathbb{S}^{d-1}$ | The unit $d$-dimensional hypersphere. |
| **Set Feature Encoder (SFE)** | |
| $\theta(x) = w^\top x$ | A linear projection function parameterized by $w \in \mathbb{S}^{d-1}$. |
| $P_i^\theta, O^\theta$ | 1D distributions obtained by projecting $P_i$ and $O$ using $\theta$. |
| $V_i^\theta = [w^\top z_{i,k}]_{k=1}^{\lvert S_i \rvert}$ | Sliced features of $P_i^\theta$. |
| $V_O^\theta = [w^\top z_h]_{h=1}^{H}$ | Sliced features of $O^\theta$. |
| $F_P(\cdot)$ | The Cumulative Distribution Function (CDF) of a distribution $P$. |
| $F_P^{-1}(\cdot)$ | The quantile function (inverse CDF) of a distribution $P$. |
| $\tau(x^\theta \mid V_O^\theta)$ | The rank of a projected value $x^\theta$ within the sorted values of $V_O^\theta$. |
| $g^+(x^\theta \mid V_i^\theta)$ | The optimal transport solver from reference slice $O^\theta$ to input slice $P_i^\theta$. |
| $\text{SFE}(V_i, V_O \mid \Theta)$ | The Set Feature Encoder function. |
| **Adversarial Perturbation and Optimization** | |
| $P_i', P_i''$ | Two perturbed distributions generated from the input set $S_i$. |
| $V_i'$ | Perturbed feature matrix of set $S_i$. |
| $v_i', v_i''$ | A pair of perturbed set embeddings for set $S_i$. |
| $\epsilon_{i,k}'$ | Random noise vector added to an element embedding. |
| $\sigma$ | The adversarial perturbation increment vector. |
| $\hat{\sigma}$ | The initial adversarial perturbation before projection. |
| $g_\sigma$ | The gradient of the loss with respect to the perturbation. |
| $v_i^\sigma$ | The final adversarially perturbed set embedding. |
| **Losses and Hyperparameters** | |
| $L_{\text{Main}}$ | The main loss for the downstream task. |
| $L_{\text{Aux}}$ | The auxiliary loss from our framework. |
| $L_{\text{wd}}$ | The InfoNCE-based loss for self-supervised learning. |
| $\Xi$ | The set of all trainable model parameters. |
| $R, H$ | Number of distribution slices and feature size of reference distribution. |
| $\pi$ | The norm constraint (radius) for perturbations. |
| $\eta, \beta$ | Step size for inner maximization and learning rate for outer minimization. |
| $\psi$ | The temperature hyperparameter for the InfoNCE loss. |
| $\lambda_1, \lambda_2$ | Hyperparameters to balance the main loss, auxiliary loss, and regularizer. |

**Remark 1.** *Let $P_i', P_i''$ denote two perturbed distributions for input set $S_i$, yielding perturbed embeddings $v_i', v_i''$. Let $S$ be the set database, $\psi$ a hyperparameter, and $SD_2$ the 2-Sliced-Wasserstein distance. For all $S_i \in S$,*

$$\mathbb{E}\left[ \frac{\exp(-\lVert v_i' - v_i'' \rVert^2 / \psi)}{\sum_{S_j \in S} \exp(-\lVert v_i' - v_j'' \rVert^2 / \psi)} \right] = \frac{\exp(-\lVert SD_2(P_i', P_i'') \rVert^2 / \psi)}{\sum_{S_j \in S} \exp(-\lVert SD_2(P_i', P_j'') \rVert^2 / \psi)}. \tag{7}$$

*Proof of Remark 1.* Let $v_j^{\theta_r}$ denote the intermediate embedding derived by the function $\theta_r$. $\| \cdot \|$ denotes the concatenation operation. Firstly, after setting $\alpha = 2$, we have:

$$\|v_i' - v_j''\|_2 \propto \left\| \frac{1}{R} \sum_{r=1}^{R} \frac{1}{H} \sum_{h=1}^{H} \left( g^+(w_r^\top z_h \mid V_i^{\theta_r'}) - g^+(w_r^\top z_h \mid V_j^{\theta_r''}) \right) \right\|_2^2$$

$$= \left\| \frac{1}{R} \sum_{r=1}^{R} \frac{1}{H} \sum_{h=1}^{H} \left( F_{P_i'^{\theta_r}}^{-1}(F_{O^{\theta_r}}(w_r^\top z_h)) - F_{P_j''^{\theta_r}}^{-1}(F_{O^{\theta_r}}(w_r^\top z_h)) \right) \right\|_2^2. \quad (8)$$

The inner summation is substituted by recognizing the relationship between the sample mean and the integral for an empirical distribution. Given an empirical distribution $P(x) = \frac{1}{H} \sum_{h=1}^{H} \delta(x \geq X_h)$, the sample mean equals the integral: $\frac{1}{H} \sum_{h=1}^{H} f(X_h) = \int f(x)\, dP(x)$. Then we apply this to the inner summation, with $t = w_r^\top z_h$ being a sample from $O^{\theta_r}$:

$$\frac{1}{H} \sum_{h=1}^{H} \left( F_{P_i'^{\theta_r}}^{-1}(F_{O^{\theta_r}}(t)) - F_{P_j''^{\theta_r}}^{-1}(F_{O^{\theta_r}}(t)) \right)^2 \quad (9)$$

$$= \int \left( F_{P_i'^{\theta_r}}^{-1}(F_{O^{\theta_r}}(t)) - F_{P_j''^{\theta_r}}^{-1}(F_{O^{\theta_r}}(t)) \right)^2 dO^{\theta_r}(t). \quad (10)$$

For the outer summation, as $R \to \infty$, we apply the Law of Large Numbers for integration over the hypersphere: $\frac{1}{R} \sum_{r=1}^{R} f(\theta_r) = \int_{\mathbb{S}^{d-1}} f(\theta)\, d\theta$. This indicates that the continuous form in expectation of above equation can be derived as:

$$= \left( \int_{\mathbb{S}^{d-1}} \int_{\mathbb{R}} \left( F_{P_i'^\theta}^{-1}(F_{O^\theta}(t)) - F_{P_j''^\theta}^{-1}(F_{O^\theta}(t)) \right)^2 dO^\theta(t)\, d\theta \right)^{1/2}$$

$$= \left( \int_{\mathbb{S}^{d-1}} \int_0^1 \left( F_{P_i'^\theta}^{-1}(y) - F_{P_j''^\theta}^{-1}(y) \right)^2 dy\, d\theta \right)^{1/2}$$

$$= \left( \int_{\mathbb{S}^{d-1}} D_2\left(P_i'^\theta, P_j''^\theta\right)^2 d\theta \right)^{1/2} = SD_2(P_i', P_j''). \quad (11)$$

And the computation for $P_i'$, $P_i''$, and $P_j''$ are independent which thus complete the proofs. $\quad \square$

**Remark 2.** *Our min-max optimization objective in Eq. (10) is approximately equivalent to an implicit regularization of the* SFE*'s local Lipschitz continuity for representation stability.*

*Proof of Remark 2.* To find the perturbation $\sigma$ that maximizes the linear approximation under the constraint $\|\sigma\|_2 \leq \pi$, we align the perturbation $\sigma$ with the gradient direction. This yields the optimal perturbation as:

$$\sigma = \pi \cdot \frac{g_\sigma}{\|g_\sigma\|_2} = \pi \cdot \frac{\nabla_\epsilon L_{wd}(\Xi; \epsilon)|_{\epsilon=0}}{\|\nabla_\epsilon L_{wd}(\Xi; \epsilon)|_{\epsilon=0}\|_2}. \quad (12)$$

Substituting this $\sigma$ back into the Taylor expansion of Eq. (11) gives the approximate value of the maximized loss as follows:

$$\max_{\|\sigma\|_2 \leq \pi} L_{wd}(\Xi, \sigma) \approx L_{wd}(\Xi, 0) + \pi \cdot \|\nabla_\epsilon L_{wd}(\Xi; \epsilon)|_{\epsilon=0}\|_2. \quad (13)$$

Thus, the original min-max objective can be approximated by the following minimization problem:

$$\min_{\Xi} \left( L_{wd}(\Xi, 0) + \pi \|\nabla_\epsilon L_{wd}(\Xi; 0)\|_2 \right). \quad (14)$$

This formulation explicitly shows that the adversarial objective encourages the minimization of not only the standard loss $L_{wd}$, but also penalizes the norm of the loss's gradient $\|\nabla_\epsilon L_{wd}(\Xi; 0)\|_2$. This second term encourages the model to be less sensitive to input perturbations.

As for $\nabla_\epsilon L_{wd}$ term, the loss $L_{wd}$ is essentially a function of the set embeddings, e.g., $v_i$, which depend on the perturbation $\epsilon$ via the SFE module: $v_i(\epsilon) = \text{SFE}(V' + \epsilon, V_O)$. We can apply the

Table 2: Hyper-parameter settings.

| | Task 1 | | Task 2 | | Task 3 | | Task 4 | | |
| | Friendster | LIVEJ | Youshu | NetEase | MLP | ISAB | LDA-1k | LDA-3k | LDA-5k |
|---|---|---|---|---|---|---|---|---|---|
| $d$ | 128 | 128 | 32 | 32 | 256 | 256 | 128 | 128 | 128 |
| $H$ | 128 | 128 | 32 | 32 | 1024 | 1024 | 128 | 128 | 128 |
| $R$ | 32 | 32 | 64 | 128 | 256 | 256 | 32 | 32 | 32 |
| $\pi$ | $5 \cdot 10^{-2}$ | $1 \cdot 10^{-1}$ | $5 \cdot 10^{-1}$ | $1 \cdot 10^{-1}$ | $1 \cdot 10^{-1}$ | $1 \cdot 10^{-2}$ | $5 \cdot 10^{-1}$ | $5 \cdot 10^{-1}$ | $5 \cdot 10^{-1}$ |
| $\psi$ | $1 \cdot 10^{-1}$ | $1 \cdot 10^{-1}$ | $5 \cdot 10^{-2}$ | $1 \cdot 10^{-1}$ | $5 \cdot 10^{-1}$ | $5 \cdot 10^{-1}$ | $1 \cdot 10^{-1}$ | $1 \cdot 10^{-1}$ | $1 \cdot 10^{-1}$ |
| $\eta$ | $1 \cdot 10^{-3}$ | $1 \cdot 10^{-2}$ | $1 \cdot 10^{-1}$ | $1 \cdot 10^{-1}$ | $1 \cdot 10^{-1}$ | $1 \cdot 10^{-2}$ | $1 \cdot 10^{-1}$ | $1 \cdot 10^{-1}$ | $1 \cdot 10^{-1}$ |
| $\beta$ | $5 \cdot 10^{-3}$ | $1 \cdot 10^{-2}$ | $1 \cdot 10^{-3}$ | $1 \cdot 10^{-3}$ | $1 \cdot 10^{-3}$ | $1 \cdot 10^{-3}$ | $1 \cdot 10^{-3}$ | $1 \cdot 10^{-3}$ | $5 \cdot 10^{-3}$ |
| $\lambda_1$ | $1 \cdot 10^{-3}$ | $1 \cdot 10^{-3}$ | $1 \cdot 10^{-2}$ | $1 \cdot 10^{-1}$ | $1 \cdot 10^{-3}$ | $1 \cdot 10^{-2}$ | $5 \cdot 10^{-1}$ | $5 \cdot 10^{-1}$ | $5 \cdot 10^{-1}$ |
| $\lambda_2$ | $1 \cdot 10^{-4}$ | $1 \cdot 10^{-4}$ | $1 \cdot 10^{-5}$ | $1 \cdot 10^{-4}$ | $1 \cdot 10^{-5}$ | $1 \cdot 10^{-5}$ | $1 \cdot 10^{-4}$ | $1 \cdot 10^{-4}$ | $1 \cdot 10^{-4}$ |

multivariable chain rule to decompose the gradient. For simplicity, we consider the gradient's dependence on a single embedding $v_i(\epsilon)$. The total derivative of $L_{wd}$ with respect to $\epsilon$, evaluated at $\epsilon = 0$, can be expressed as:

$$\nabla_\epsilon L_{wd}(\Xi; 0) = \sum_{i \in \mathcal{B}_{\text{Batch}}} \left( J_{\text{SFE}}(V_i') \right)^\top \nabla_{v_i} L_{wd}. \tag{15}$$

Here $\nabla_{v_i} L_{wd}$ is the loss gradient with respect to the vector embedding $v_i$, and $J_{\text{SFE}}(V_i')$ is the Jacobian matrix of the SFE function with respect to its input features. The regularizer in Eq. (13) penalizes the norm of this sum. For this norm to be small, the model is incentivized to reduce the norms of its constituent components, most notably the spectral norm of the SFE Jacobian, $\|J_{\text{SFE}}\|_2$.

This penalty on the Jacobian norm is directly related to the local Lipschitz continuity of the SFE function. For a differentiable function, its local Lipschitz constant $L$ over a region $\mathcal{V}$ is bounded by the supremum of the spectral norm of its Jacobian within that region:

$$L(\text{SFE}, \mathcal{V}) = \sup_{V \in \mathcal{V}} \|J_{\text{SFE}}(V)\|_2. \tag{16}$$

By encouraging a smaller Jacobian norm, our adversarial optimization implicitly regularizes the SFE to have a smaller local Lipschitz constant. A smaller constant ensures that for any two nearby feature sets $V_i$ and $V_j$, the distance between their embeddings is bounded:

$$\|\text{SFE}(V_i) - \text{SFE}(V_j)\|_2 \leq L(\text{SFE}) \cdot \|V_i - V_j\|_2. \tag{17}$$

This property means that small, non-semantic perturbations to the input features will only result in small, bounded changes to the output embedding, which defines the representation stability (Donchev and Farkhi, 1998). □

# E SUPPLEMENTARY DETAILS OF EXPERIMENTS

## E.1 HYPER-PARAMETER SETTINGS

We report all hyper-parameter settings in Table 2.

## E.2 EXPERIMENT CONFIGURATIONS

We implement all models by Python 3.8 and PyTorch 1.12.0 with non-distributed training. We run on a Linux machine with 4 NVIDIA A100-PCIE-40GB GPUs and 10 vCPU Intel Xeon Processor (Skylake, IBRS). We adhere to the officially reported hyperparameter settings of all baselines and conduct a grid search to models without prescribed configurations. The learning rate is tuned in the range $\{10^{-4}, 10^{-3}, 10^{-2}\}$. Optimization for all models is performed using the default Adam optimizer (Kingma and Ba, 2015). All results are averaged based on five-fold evaluations.

## E.3 TASK 1: LEARNING TO RANK SET SIMILARITY

### E.3.1 DATASET DESCRIPTIONS

We incorporate two large real-world datasets, i.e., Friendster (Yang and Leskovec, 2015) and LIVEJ (Mislove et al., 2007), for set similarity learning evaluation. We filter out sets with fewer than three

Table 3: Similarity metrics.

| Jaccard | Cosine | NOverlap | Dice |
|---------|--------|----------|------|
| $\frac{|s_1 \cap s_2|}{|s_1 \cup s_2|}$ | $\frac{|s_1 \cap s_2|}{\sqrt{|s_1||s_2|}}$ | $\frac{|s_1 \cap s_2|}{\max(|s_1|,|s_2|)}$ | $\frac{2|s_1 \cap s_2|}{|s_1|+|s_2|}$ |

Table 4: Dataset statistics of Task 1.

| Dataset | #S | #E | # Avg. E/S |
|---------|------|------|------|
| Friendster | 889,839 | 5,501,401 | 11.29 |
| LIVEJ | 1,205,816 | 1,975,812 | 9.90 |

elements. Specifically, Friendster is the dataset from an online social gaming site[1] where each set corresponds to a group membership. LIVEJ is a dataset from a free online community LiveJournal[2].

With coefficients $w_1-w_4$ randomly sampled from $(0,1)$, we incorporate four widely-used similarity metrics, i.e., Jaccard, Cosine, Normalized Overlap (denoted as NOverlap), and Dice, as the score functions to construct similar sets. Their formulations are reported in Table 3. Given a set, e.g., $s_i$, its similarity value $\mathcal{M}$ to another set, e.g., $s_j$, is computed as:

$$\mathcal{M}(s_i, s_j) = w_1 \mathcal{M}_{\text{Jaccard}}(s_i, s_j) + w_2 \mathcal{M}_{\text{Cosine}}(s_i, s_j) + \\ w_3 \mathcal{M}_{\text{NOverlap}}(s_i, s_j) + w_4 \mathcal{M}_{\text{Dice}}(s_i, s_j). \tag{18}$$

This construction process ensures that the evaluation is free from selection bias and captures a wide spectrum of possible similar set configurations. With #S and #E denoting the numbers of sets and elements, the result data statistics are reported in Table 4.

### E.3.2 METHOD DESCRIPTIONS

The methods for set similarity learning are introduced as follows:

- **SAP** (Lin et al., 2013) denotes the classic implementation with global average pooling methodology for sets.
- **SMP** (Lin et al., 2013) is the implementation with set max pooling.
- **DeepSet** (Zaheer et al., 2017) is another classic set embedding method that learns permutation-invariant functions with deep neural networks. We implement global mean pooling as the permutation-invariant function of DeepSet.
- **RepSet** (Skianis et al., 2020) extracts set representations by computing the bipartite matching costs between the input set and a collection of learnable reference sets.
- **SAtt** (Lee et al., 2019) is a state-of-the-art method to embed set structures with self-attention mechanism and Transformer architecture.
- **PoT** (Guo et al., 2021a) introduces a prototype-oriented optimal transport framework that learns set representations by minimizing the transport distance between the set's empirical distribution and learnable global prototypes.
- **Set2Box** (Lee et al., 2022) maps sets into hyper-rectangular box embeddings to effectively capture logical relationships and set boundaries.
- **OTKE** (Mialon et al., 2021) introduces a trainable embedding scheme based on kernelized optimal transport to aggregate set features.
- **DIEM** (Kim, 2022) proposes a differentiable framework to learn informative set interactions and enhance representation distinctiveness.
- **FSPool** (Zhang et al., 2020) is a representative set embedding framework with carefully-designed deep learning architecture.
- **PSWE** (Naderializadeh et al., 2021) is one of the state-of-the-art deep learning model for set representation learning.

---

[1] http://www.friendster.com/
[2] http://www.livejournal.com/

Table 5: Detailed performance comparison for Task 1.

| | **Task 1: Set Similarity Learning** | | | | | | | |
|---|---|---|---|---|---|---|---|---|
| | Friendster | | | | LIVEJ | | | |
| Model | R@20 | N@20 | R@100 | N@100 | R@20 | N@20 | R@100 | N@100 |
| SAP | $72.41 \pm 0.18$ | $68.13 \pm 0.29$ | $85.32 \pm 0.15$ | $72.15 \pm 0.26$ | $79.86 \pm 0.11$ | $77.75 \pm 0.14$ | $86.94 \pm 0.09$ | $85.34 \pm 0.12$ |
| SMP | $70.78 \pm 0.28$ | $68.99 \pm 0.30$ | $81.61 \pm 0.11$ | $72.22 \pm 0.25$ | $78.95 \pm 0.12$ | $76.70 \pm 0.08$ | $86.83 \pm 0.10$ | $84.65 \pm 0.09$ |
| DeepSet | $63.20 \pm 0.67$ | $60.75 \pm 0.69$ | $76.60 \pm 1.64$ | $69.89 \pm 0.60$ | $75.45 \pm 0.09$ | $74.55 \pm 0.11$ | $83.31 \pm 0.05$ | $79.76 \pm 0.06$ |
| RepSet | $80.63 \pm 0.19$ | $76.56 \pm 0.21$ | $86.49 \pm 0.14$ | $74.92 \pm 0.23$ | $82.15 \pm 0.18$ | $79.63 \pm 0.22$ | $88.41 \pm 0.15$ | $83.12 \pm 0.19$ |
| SAtt | $77.52 \pm 0.45$ | $71.92 \pm 0.47$ | $87.51 \pm 0.32$ | $75.21 \pm 0.42$ | $83.79 \pm 0.06$ | $81.73 \pm 0.11$ | $91.39 \pm 0.02$ | $85.07 \pm 0.10$ |
| PoT | $82.44 \pm 0.12$ | $81.85 \pm 0.16$ | $86.96 \pm 0.15$ | $81.47 \pm 0.18$ | $83.18 \pm 0.14$ | $84.25 \pm 0.17$ | $89.33 \pm 0.11$ | $86.45 \pm 0.13$ |
| Set2Box | $67.35 \pm 0.22$ | $69.73 \pm 0.25$ | $73.46 \pm 0.18$ | $70.33 \pm 0.20$ | $77.24 \pm 0.20$ | $75.89 \pm 0.23$ | $85.12 \pm 0.16$ | $82.34 \pm 0.21$ |
| OTKE | $79.53 \pm 0.14$ | $73.68 \pm 0.21$ | $86.64 \pm 0.11$ | $79.59 \pm 0.17$ | $81.45 \pm 0.16$ | $79.82 \pm 0.19$ | $87.95 \pm 0.13$ | $85.10 \pm 0.15$ |
| DIEM | $82.49 \pm 0.16$ | $81.40 \pm 0.13$ | $88.36 \pm 0.09$ | $81.56 \pm 0.15$ | $83.95 \pm 0.12$ | $84.92 \pm 0.15$ | $89.88 \pm 0.08$ | $87.15 \pm 0.11$ |
| PSWE | $83.05 \pm 0.09$ | $84.26 \pm 0.15$ | $88.59 \pm 0.05$ | $85.77 \pm 0.14$ | $83.52 \pm 0.09$ | $84.61 \pm 0.13$ | $89.48 \pm 0.02$ | $86.67 \pm 0.09$ |
| FSPool | $79.90 \pm 0.13$ | $81.96 \pm 0.15$ | $87.76 \pm 0.11$ | $84.41 \pm 0.14$ | $85.36 \pm 0.03$ | $87.17 \pm 0.08$ | $93.07 \pm 0.06$ | $90.29 \pm 0.06$ |
| FSW | $83.58 \pm 0.13$ | $84.39 \pm 0.10$ | $88.52 \pm 0.07$ | $85.81 \pm 0.12$ | $84.19 \pm 0.05$ | $85.04 \pm 0.06$ | $89.95 \pm 0.09$ | $87.23 \pm 0.10$ |
| **SRAL** | $91.57 \pm 0.22$ | $92.22 \pm 0.22$ | $94.53 \pm 0.11$ | $93.01 \pm 0.19$ | $87.56 \pm 0.31$ | $89.31 \pm 0.31$ | $92.93 \pm 0.02$ | $91.25 \pm 0.02$ |
| Gain | $9.56\%^*$ | $9.28\%^*$ | $6.71\%^*$ | $8.39\%^*$ | $2.58\%^*$ | $2.46\%^*$ | $-0.15\%$ | $1.06\%^*$ |

Table 6: Data statistics of Task 2. S, U, and E denote sets (bundles), users, and elements.

| Dataset | # S | # U | # E | # Avg. E/S |
|---|---|---|---|---|
| Youshu | 4,771 | 8,039 | 32,770 | 37.03 |
| NetEase | 22,864 | 18,528 | 123,623 | 77.80 |

- **FSW** (Amir and Dym, 2025) is another state-of-the-art model that utilizes the Fourier transform in the frequency domain for "multisets".

### E.3.3   DETAILED EXPERIMENTAL RESULTS

For completeness, we present the detailed experimental results for Task 1 in Table 5. These results including mean scores and standard deviations over multiple runs, substantiate the findings discussed in the main text, where our model SRAL consistently outperforms all baselines.

### E.4   TASK 2: BUNDLE REPRESENTATION LEARNING FOR RECOMMENDATION

### E.4.1   DATASET DESCRIPTIONS

Following recent works (Ma et al., 2022; Chang et al., 2020; Deng et al., 2020), we include two real-world datasets: Youshu (Chen et al., 2019) for book list recommendation and NetEase (Cao et al., 2017) for music playlist recommendation. Dataset statistics are reported in Table 6.

### E.4.2   METHOD DESCRIPTIONS

We include the following bundle recommender models:

- **MFBPR** (Rendle et al., 2012) utilizes the Bayesian Personalized Ranking (BPR) loss within a Matrix Factorization framework to model collaborative filtering between users and bundles.

- **DSBRec** is a specialized implementation with DeepSet (Zaheer et al., 2017) and optimize with BPR loss.

- **DAM** (Chen et al., 2019) attentively captures bundle representations from associated items and utilizes multi-task learning to optimize interactions between users and both items and bundles.

- **BundleNet** (Deng et al., 2020) is a traditional framework for bundle recommendation that constructs a tripartite graph of user-bundle-element relationships and utilizes Graph Convolution Network (GCN) for representation learning alongside multi-task learning.

- **BGCN** (Chang et al., 2020), a bundle recommendation method, dissects user-bundle-element relationships into two distinct perspectives: bundle-view and item-view graphs.

- **CrossCBR** (Ma et al., 2022) leverages contrastive learning to achieve cross-view alignment in the latent space.

Table 7: Detailed performance comparison for Task 2.

| | Youshu | | | | NetEase | | | |
|---|---|---|---|---|---|---|---|---|
| **Task 2: Bundle Recommendation** | | | | | | | | |
| Model | R@20 | N@20 | R@100 | N@100 | R@20 | N@20 | R@100 | N@100 |
| MFBPR | $19.97 \pm 0.45$ | $11.67 \pm 0.52$ | $44.33 \pm 0.33$ | $17.95 \pm 0.41$ | $5.21 \pm 0.15$ | $2.98 \pm 0.11$ | $14.15 \pm 0.18$ | $4.92 \pm 0.12$ |
| DSBRec | $20.46 \pm 0.41$ | $12.03 \pm 0.31$ | $45.34 \pm 0.32$ | $18.12 \pm 0.28$ | $5.51 \pm 0.10$ | $3.04 \pm 0.08$ | $14.76 \pm 0.13$ | $5.14 \pm 0.14$ |
| DAM | $20.83 \pm 0.36$ | $11.99 \pm 0.22$ | $45.58 \pm 0.26$ | $18.38 \pm 0.33$ | $5.54 \pm 0.13$ | $3.11 \pm 0.09$ | $14.98 \pm 0.16$ | $5.12 \pm 0.10$ |
| BundleNet | $22.85 \pm 0.35$ | $11.90 \pm 0.25$ | $47.84 \pm 0.22$ | $19.19 \pm 0.31$ | $6.17 \pm 0.12$ | $3.44 \pm 0.07$ | $16.26 \pm 0.11$ | $5.83 \pm 0.08$ |
| BGCN | $25.22 \pm 0.12$ | $14.54 \pm 0.10$ | $49.38 \pm 0.26$ | $21.18 \pm 0.29$ | $7.04 \pm 0.10$ | $3.91 \pm 0.08$ | $17.25 \pm 0.15$ | $6.51 \pm 0.09$ |
| CrossCBR | $\mathbf{26.41 \pm 0.42}$ | $\mathbf{16.55 \pm 0.23}$ | $\mathbf{51.90 \pm 0.44}$ | $\mathbf{23.30 \pm 0.25}$ | $\mathbf{7.21 \pm 0.11}$ | $\mathbf{4.08 \pm 0.06}$ | $\mathbf{18.32 \pm 0.13}$ | $\mathbf{6.77 \pm 0.07}$ |
| SRAL$^+$ | $\mathbf{26.92 \pm 0.09}$ | $\mathbf{16.95 \pm 0.08}$ | $\mathbf{52.18 \pm 0.19}$ | $\mathbf{23.64 \pm 0.06}$ | $\mathbf{7.37 \pm 0.11}$ | $\mathbf{4.21 \pm 0.05}$ | $\mathbf{18.66 \pm 0.01}$ | $\mathbf{7.01 \pm 0.06}$ |
| Gain | 1.93%* | 2.42%* | 0.54%* | 1.46%* | 2.22%* | 3.19%* | 1.86%* | 3.54%* |

Table 8: Test accuracy (%) of SRAL and competing methods for Task 3.

| | | | | | **Task 3: Point Cloud Processing** | | | | | | | | |
|---|---|---|---|---|---|---|---|---|---|---|---|---|---|
| Backbone | SAP | SMP | RepSet | SAtt | PoT | Set2Box | OTKE | DIEM | PSWE | FSPool | FSW | SRAL | Gain |
| MLP | $57.65 \pm 0.52$ | $86.35 \pm 0.43$ | $83.45 \pm 0.55$ | $85.89 \pm 0.41$ | $85.20 \pm 0.48$ | $82.15 \pm 0.64$ | $85.92 \pm 0.38$ | $85.58 \pm 0.42$ | $86.41 \pm 0.39$ | $85.76 \pm 0.32$ | $\mathbf{86.38 \pm 0.35}$ | $\mathbf{86.53 \pm 0.36}$ | +0.14% |
| ISAB | $85.45 \pm 0.16$ | $86.82 \pm 0.49$ | $86.05 \pm 0.31$ | $86.78 \pm 0.28$ | $86.55 \pm 0.25$ | $85.88 \pm 0.34$ | $86.70 \pm 0.29$ | $86.72 \pm 0.26$ | $86.85 \pm 0.30$ | $86.88 \pm 0.53$ | $\mathbf{86.93 \pm 0.21}$ | $\mathbf{87.31 \pm 0.23}$ | +0.44%* |

### E.4.3 DETAILED EXPERIMENTAL RESULTS

Table 7 catalogs the precise outcomes on both the Youshu and NetEase datasets, with standard deviations included to underscore the stability of our findings. The results show that SRAL$^+$ consistently outperforms other rival methods with low standard deviations across all runs.

### E.5 TASK 3: POINT CLOUD PROCESSING

#### E.5.1 DATASET DESCRIPTIONS

For this study, we utilize ModelNet40 dataset Wu et al. (2015), comprising 3D point clouds extracted from triangular meshes of 12,311 computer-aided design models across 40 distinct object categories. Each object is represented by a set of 1024 points, following methodologies outlined in Guo et al. (2021b); Qi et al. (2017).

#### E.5.2 DETAILED EXPERIMENTAL RESULTS

For Task 3, Table 8 summarizes the comparative test accuracies using both MLP and ISAB backbones. The inclusion of standard deviations further illustrates the stability of these outcomes.

### E.6 TASK 4: TOPIC SET EXPANSION

#### E.6.1 DATASET DESCRIPTIONS

We leverage three datasets[3], i.e., LDA-1k, LDA-3k, and LDA-5k, from previous work Zaheer et al. (2017) that are originally processed from latent Dirichlet allocation Blei et al. (2003). They[4] respectively contain 2,000, 6,000, and 10,000 sets and 17,016, 37,718 and 61,127 vocabulary elements. Their average elements per set are around 25.

#### E.6.2 DETAILED EXPERIMENTAL RESULTS

The complete results with standard deviations of Task 4 evaluation are reported in Table 9.

### E.7 IMPLEMENTATION OF SELF-SUPERVISED LEARNING LOSS

We implement the following self-supervised learning losses for comparison:

- **Set Triplet Loss:**

---

[3] https://github.com/mysbupt/CrossCBR/blob/master/dataset.tgz
[4] https://github.com/manzilzaheer/DeepSets/tree/master/SetExpansion/data/lda

Table 9: AUC results (%) of SRAL and competing methods for Task 4.

| | | | | | | Task 4: Topic Set Expansion | | | | | | | | |
|---|---|---|---|---|---|---|---|---|---|---|---|---|---|---|
| Data | SAP | SMP | DeepSet | RepSet | SAtt | PoT | Set2Box | OTKE | DIEM | PSWE | FSPool | FSW | SRAL | Gains |
| LDA-1k | $54.34 \pm 3.91$ | $67.21 \pm 6.69$ | $54.98 \pm 5.11$ | $57.32 \pm 2.34$ | $58.55 \pm 2.70$ | $58.94 \pm 3.12$ | $50.59 \pm 4.05$ | $62.95 \pm 3.88$ | $63.58 \pm 3.56$ | $58.36 \pm 5.49$ | $\mathbf{75.67 \pm 4.01}$ | $64.56 \pm 3.41$ | $\mathbf{80.94 \pm 1.38}$ | +6.96%* |
| LDA-3k | $51.95 \pm 1.87$ | $74.40 \pm 2.12$ | $51.96 \pm 3.91$ | $58.33 \pm 1.20$ | $77.48 \pm 3.84$ | $73.40 \pm 2.45$ | $64.98 \pm 3.22$ | $77.59 \pm 2.91$ | $75.67 \pm 2.15$ | $78.44 \pm 2.04$ | $70.57 \pm 1.38$ | $\mathbf{79.67 \pm 2.35}$ | $\mathbf{87.93 \pm 1.92}$ | +10.37%* |
| LDA-5k | $51.34 \pm 1.34$ | $80.65 \pm 1.18$ | $52.05 \pm 1.28$ | $61.39 \pm 2.31$ | $74.59 \pm 3.37$ | $75.11 \pm 2.08$ | $65.67 \pm 2.54$ | $72.57 \pm 2.76$ | $76.96 \pm 2.33$ | $78.81 \pm 2.92$ | $71.16 \pm 1.78$ | $\mathbf{80.94 \pm 2.34}$ | $\mathbf{86.20 \pm 0.67}$ | +6.50%* |

Table 10: Runtime comparison for Task 1 and Task 2.

| Task 1: Set Similarity Learning Runtime | | | | | | Task 2: Bundle Recommendation Runtime | | | | |
|---|---|---|---|---|---|---|---|---|---|---|
| | Friendster | | LIVEJ | | | | Youshu | | NetEase | |
| Model | In Total | Per-epoch | In Total | Per-epoch | | Model | In Total | Per-epoch | In Total | Per-epoch |
| PSWE | 3.1h | 1.6min | 5.1h | 2.7min | | BGCN | 12.6min | 6.3s | 23.4min | 9.7s |
| FSPool | 2.9h | 1.5min | 6.5h | 2.1min | | CrossCBR | 10.4min | 4.5s | 19.5min | 8.4s |
| SRAL | 3.9h | 5.6min | 7.7h | 7.9min | | SRAL$^+$ | 24.5min | 42.2s | 44.8min | 57.4s |

$$\mathcal{L} = \sum_{S_i, S_j \in S} \max\big(d(v_i, v_i') - d(v_i, v_j') + \alpha, 0\big). \tag{19}$$

where $\alpha$ is the hyperparameter.

- **Soft-Nearest Neighbors Loss:**

$$\mathcal{L} = -\log \frac{\exp(\mathrm{sim}(v_i', v_i'')/\tau)}{\sum_{j \neq i} \exp(\mathrm{sim}(v_i', v_j')/\tau)} - \log \frac{\exp(\mathrm{sim}(v_i'', v_i')/\tau)}{\sum_{j \neq i} \exp(\mathrm{sim}(v_i'', v_j'')/\tau)}. \tag{20}$$

where we use inverse Euclidean distance to implement the $\mathrm{sim}$ function. For this experiment, it is important to note that these metrics cannot be directly integrated into our SFE architecture. Therefore, we made adaptations for them as follows: for sets of varying sizes, we first aggregate their element embeddings using mean pooling to obtain a single vector representation for each set. Subsequently, we employ each respective metric to calculate the distributional distance. The training objective is to align these distances with the ground-truth similarity rankings from Task 1.

- **Barlow Twins Loss:**

$$\mathcal{L} = \sum_{k=1}^{D}(1 - C_{kk})^2 + \lambda \sum_{k=1}^{D} \sum_{\substack{l=1 \\ l \neq k}}^{D} (C_{kl})^2, \tag{21}$$

where

$$C = \frac{1}{\mathrm{BatchSize}} \mathrm{norm}(V')^\top \mathrm{norm}(V''). \tag{22}$$

Here $V'$ and $V''$ denote the perturbed batch set embeddings.

### E.8    SCALABILITY STUDY OF SRAL

To assess the scalability of SRAL, we derived nine proportionally-sized sub-datasets from Friendster. Figure 1 presents a scalability analysis of SRAL on subsets of the Friendster dataset. We have two main observations. First, the training time per epoch (blue bars, left axis) exhibits a gradually growing trend as the data volume increases. This prevents prohibitive computational costs on larger datasets and confirms the model's acceptable efficiency. Second, the model's performance, measured by Recall@20 (orange line, right axis), remains generally consistent with slight fluctuation. These results demonstrate the effectiveness of our SRAL model, making it a practical solution for large-scale settings.

### E.9    COMPUTATION EFFICIENCY ANALYSIS

### E.9.1    COMPUTATION COST COMPARISON

We report the training time cost of SRAL and compare it with two most competitively performing models across all tasks. As shown in Tables 10 and 11, although our SRAL model incurs a higher

Table 11: Runtime comparison for Task 3 and Task 4.

| **Task 3: Point Cloud Processing Runtime** | | | | | | **Task 4: Topic Set Expansion Runtime** | | | | | | |
|---|---|---|---|---|---|---|---|---|---|---|---|---|
| | MLP | | ISAB | | | | LDA-1k | | LDA-3k | | LDA-5k | |
| Model | In Total | Per-epoch | In Total | Per-epoch | | Model | In Total | Per-epoch | In Total | Per-epoch | In Total | Per-epoch |
| PSWE | 1.6h | 27.4s | 1.8h | 29.5s | | PSWE | 10.5s | 0.3s | 15.6s | 0.3s | 18.5s | 0.3s |
| FSPool | 1.2h | 23.9s | 1.6h | 26.3s | | FSPool | 9.8s | 0.2s | 14.4s | 0.2s | 17.4s | 0.3s |
| SRAL | 2.3h | 1.6min | 2.7h | 1.7min | | SRAL | 14.3s | 0.5s | 23.5s | 0.5s | 31.6s | 0.6s |

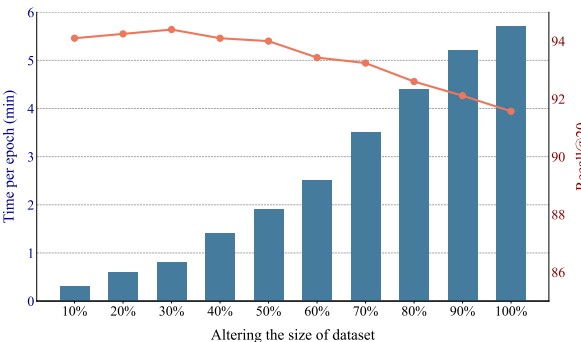

Figure 1: Results with varying data sizes.

"Per-epoch" computational cost compared to the baselines, its "In Total" training time remains comparable. This is because our proposed Adversarial Encoding Perturbation and Optimization mechanism well promotes the model convergence, as we analyzed earlier in Section 4.3.2.

### E.9.2    MODULE COMPUTATION COST

To offer deeper insights into the computational overhead, we break down the per-epoch training time for individual modules using the Friendster dataset from Task 1 as follows:

As shown in Table 12, the Adversarial Encoding Perturbation and Optimization (AEPO) module incurs the highest computational cost among all components. This overhead primarily arises from the adversarial perturbation generation and the iterative min-max optimization process. Nevertheless, considering its substantial contributions to accelerating convergence and significantly improving training stability, we believe that this computational expenditure constitutes a worthwhile trade-off.

### E.10    MODULE COMPUTATION COST

To offer deeper insights into the computational overhead, we break down the per-epoch training time for individual modules using the Friendster dataset from Task 1 as follows:

As shown in Table 12, the Adversarial Encoding Perturbation and Optimization (AEPO) module incurs the highest computational cost among all components. This overhead primarily arises from the adversarial perturbation generation and the iterative min-max optimization process. Nevertheless, considering its substantial contribution to accelerating convergence and significantly improving training stability, we believe that this computational expenditure constitutes a worthwhile trade-off.

### E.11    COMPARISON WITH VANILLA OT-BASED SOLUTIONS

While Optimal Transport (OT) has been explored in set representation learning, e.g., *RepSet* Skianis et al. (2020), our approach distinguishes itself through the utilization of the Sliced-Wasserstein (SW) distance and a tailored adversarial enhancement mechanism. *RepSet* formulates the set distance as a bipartite matching problem, which is equivalent to exact OT, and offers an approximation variant (*ApproxRepSet*) by relaxing constraints to a semi-relaxed OT problem. To provide the comparison, we conducted experiments on the Friendster dataset (Task 1) covering three aspects: (1) direct performance comparison against *RepSet* and *ApproxRepSet*; (2) evaluating *RepSet* variants as encoders within our SRAL framework. The results are summarized in Table 13.

Table 12: Training time cost of each major module.

|  | SRAL (complete) | SRAL (SFE Module) | SRAL (AEPO) | SRAL (Main Loss) |
|---|---|---|---|---|
| Youshu | 5.6min | 1.8min | 3.3min | 0.1min |

Table 13: Performance comparison with RepSet and its variants.

|  | RepSet | SRAL (RepSet) | ApproxRepSet | SRAL (ApproxRepSet) | SRAL |
|---|---|---|---|---|---|
| Recall@20 | 80.63 | 82.25 | 77.81 | 78.93 | **91.57** |

**Performance Analysis.** We observe that our SRAL outperforms the vanilla *RepSet* baseline; and the approximation variant *ApproxRepSet* exhibits inferior performance. We attribute this gap to the nature of the "semi-relaxed" approximation employed in *ApproxRepSet*. As noted in Skianis et al. (2020), dropping constraints to achieve computational efficiency may sacrifice rigorous metric properties, such as the triangle inequality, and makes the optimization prone to local optima. In contrast, our SW-based approach preserves key geometric properties while remaining computationally efficient, resulting in superior representation stability.

**Encoder Compatibility.** When integrating *RepSet* as the encoder within our framework, we observe a performance gain but still falls short of our native SRAL model. This empirical finding validates our theoretical analysis in Remark 1: our AEPO mechanism is specifically tailored for the Set Feature Encoder (SFE) based on Sliced-Wasserstein metric. The adversarial perturbations generated by AEPO aim to maximize discrepancies in the embedding space, which are positively correlated in expectation with the Sliced-Wasserstein distance. In contrast, such a direct perturbation may not hold for the bipartite matching objective in *RepSet*, rendering the adversarial optimization less effective in capturing distributional semantics.

