# OpenReview forum: "Adversarial Encoding Perturbation and Synthesis for Set Representation Auxiliary Learning"
_ICLR.cc/2026/Conference — ICLR 2026 Poster_

### Official Review · Reviewer_kDXr · 2025-10-29

**Soundness:** 3
**Presentation:** 1
**Contribution:** 2
**Rating:** 4
**Confidence:** 4

**Summary:**

The paper presents a new method, SRAL, for set representation learning. To encode a set, the authors first extract distributional features from it using sliced wasserstein distances rank-matching to a learned canonical set. These set representations are then inserted into a contrastive loss for learning the canonical set embeddings. To choose the positive pairs, the authors augment each set by adding noise to its elements and then extracting the new noisy set embeddings. Specifically, to increase the augmentations effect, they optimize an adversarial noise direction in an inner loop.

**Strengths:**

- While writing clarity could be improved, the proposed method appears simple and intuitive for set representation.

- The analsys presented in remark 1, showing the connection between the proposed loss metric and a sliced wasserstein based contrastive loss, is interesting.

**Weaknesses:**

- **Writing clarity of method sections.** Sections 2-3 would benefit from simplified presentation. Although the proposed method appears simple, it is hard to follow. Specifically, it is notation-heavy (the notation table in the appendix is almost a page long) which further reduces clarity. Rewriting these sections in a simplified manner can greatly improve the paper's clarity.

- **Limited baseline comparisons.** The experimental section is missing several key comparisons to more recent methods. Moreover, when SRAL is compared to a more recent method, CrossCBR, it shows only marginal gains. The paper would be strengthened by adding comparisons to [1,2,3] on tasks 1 and 4 to evaluate SRAL against more recent approaches.

- **Marginal gains in tasks 2 & 3.**  The results of SRAL on tasks 2 and 3 show only marginal gains over the baselines. Additionally, on Task 2, SRAL is integrated into CrossCBR rather than evaluated standalone. Providing standalone results for Task 2 would help clarify SRAL's direct contribution.


Minor remarks:

- **Figures and tables sizing.** Most figures and tables are quite small and difficult to read, particularly in printed form. Enlarging key results would improve readability.


[1] Lee, Dong Bok, et al. "Self-supervised set representation learning for unsupervised meta-learning." The Eleventh International Conference on Learning Representations. 2023.

[2] Guo, Dandan, et al. "Learning prototype-oriented set representations for meta-learning." arXiv preprint arXiv:2110.09140 (2021).


[3] Lee, Geon, Chanyoung Park, and Kijung Shin. "Set2box: Similarity preserving representation learning for sets." 2022 IEEE International Conference on Data Mining (ICDM). IEEE, 2022.

**Questions:**

- How does the authors view an integragtion of a powerful LLM into this problem? A discussion on that could be helpful.

- Except the embeddings of the canonical set V_o, what other parameters are learned in SRAL?

---

> ### Author Response · Authors · 2025-11-21
> **Resonse to Reviewer kdXr (Part I)**
>
> We sincerely appreciate Reviewer kDXr for your valuable feedback. We have carefully addressed each point below and hope these responses satisfactorily resolve your concerns. In the revised manuscript, we use blue to highlight the corresponding modifications.
>
> **W1**:
> We sincerely thank Reviewer kDXr for your constructive suggestions. While preserving the necessary mathematical rigor of the domain-specific formulations, we have extensively polished Sections 2 and 3 to enhance readability, including simplifying notations and adding concrete illustrative examples. Specifically
> - In Section 2 (L.105–124): We have expanded Problem Description and introduced an Illustrative Example to facilitate understanding for a wider audience.
> - Additionally, we streamlined the notation by replacing $P\_{\<g\>} \to P^\theta$ (L.130). In Section 3.2 (L.182-187): We have added detailed explanations for the end-to-end learning mechanism of the reference set embeddings.
> - In Sections 3.2 and 3.3: We have simplified notations throughout the text to improve flow, e.g., ${P\_i}\_{<\theta>} \to P\_i^\theta$, ${O}\_{<\theta>} \to O^\theta$, and $\epsilon^+ \to \sigma$. We also incorporated further explanatory text in Section 3.3 (L. 255-259, L.272-274) to clarify the optimization details.
>
> All modifications in the revised manuscript are highlighted in blue. We hope these revisions effectively address your concerns, and we remain open to any further feedback.
>
>
> **W2**:
> Thanks Reviewer kDXr for your feedback.
> - For Task 2, this is because state-of-the-art bundle recommendation model relies on Collaborative Filtering (CF) signals to handle complex `user-bundle interactions`, e.g., via GNNs, which may not be simply captured by set representation learning alone. Although performance improvement appears modest in magnitude, we notice that they are statistically significant. This demonstrates the compatibility of our method.
> - Following your suggestion, we have incorporated POT [1] into our comparison. In its original formulation, the output $h_j$ serves as a $K$-dimensional weight vector over global prototypes rather than the $d$-dimensional dense embedding required for our downstream tasks. To adapt this to our setting, we followed the strategy in [1] by utilizing the intermediate $d$-dimensional feature from the backbone as the final set representation. Regarding Set-SimCLR [2], we were unable to include it due to the lack of public codes. For Set2Box [3], which employs a "unique box embedding format" that is distinct from other methods, we thus computed the mean of its box embedding to derive a compatible set embedding for comparison. Furthermore, we introduced two recent Optimal Transport-based methods, FSW [4] and DIEM [5]. The results on Task 1 and Task 4 are presented in the table below. We notice that our method consistently outperforms these baselines, primarily attributed to our SFE and AEPO mechanisms which effectively enhance the quality of the learned set embeddings.
> We have updated the revision with discussions on all these mehtods and corresponding results in Experiment Section. Thank you.
>
>     |Task 1|PoT|Set2Box|FSW|DIEM|SRAL|
>     |----|----|----|----|----|----|
>     |R@20|82.44|67.35|83.58|82.49|91.57|
>     |R@100|86.96|73.46|88.52|88.36|94.53|
>
>      |Task 4|PoT|Set2Box|FSW|DIEM|SRAL|
>     |----|----|----|----|----|----|
>     |LDA-1k|58.94|50.59|64.56|63.58|80.94|
>     |LDA-3k|73.40|64.98|75.67|75.67|87.93|
>     |LDA-5k|75.11|65.67|80.94|76.96|86.20|
>
> [1] Learning prototype-oriented set representations for meta-learning. ICLR 2022
> [2] Self-supervised set representation learning for unsupervised meta-learning. ICLR 2023
> [3] Set2box: Similarity preserving representation learning for sets. ICDM 2022.
> [4] Fourier Sliced-Wasserstein Embedding for Multisets and Measures. ICLR 2025
> [5] Differentiable expectation-maximization for set rpresentation learning, ICLR 2023

---

> > ### Author Response · Authors · 2025-11-21
> > **Response to Reviewer kdXr (Part II)**
> >
> > **W3**:
> > We thank Reviewer kDXr for your constructive suggestion.
> >
> > -  For Task 2, as we explained earlier, CF signals can not be learned simply by set representation learning. Therefore, our strategy was to integrate SRAL into established backbones (like CrossCBR) to enhance the bundle semantic representation within the CF framework.
> > - To address your suggestion for a standalone assessment, we implemented a minimalist version where SRAL encodes bundle features and is optimized directly with user embeddings via Matrix Factorization (optmized by MFBPR), without advanced graph propagation. We compared this against the standard MFBPR baseline as follows. As shown above, SRAL can improve over the MFBPR baseline. However, the absolute performance is naturally lower than CrossCBR-based methods due to the lack of strong graph-based CF signal modeling.
> >
> >     ||MFBPR - Youshu|SRAL(MFBPR) - Youshu|MFBPR - NetEase|SRAL(MFBPR) - NetEase|
> >     |----|----|----|----|----|
> >     |R@20|19.97|20.59|5.21|5.67|
> >     |R@100|44.33|45.20|14.15|14.49|
> >
> > - For Task 3, we respectfully note that the baseline accuracy is already very high (>86\%), indicating that a large numerical improvement of performance metric may inherently not be that easy. Ultimately, we humbly emphasize that the core value of SRAL lies in its design as a general-purpose framework. It delivers balanced and statistically significant improvements across diverse tasks.
> >
> > **W4**:
> > Thank you Reviewer kDXr for the thoughtful suggestions. In revision, we have refined all figures and expanded tables, e.g., Table 2 and Table 3, to enhance readability . For other tables, we have moved the detailed tabular results to Appendix. Please kindly check and let us know if there are any remaining issues.
> >
> > **Q1**: Thanks Reviewer kDXr for your insightful question.
> > LLMs could be a promising tool to boost our SRAL framework. Specifically,
> > - LLM could be a powerful feature extractor. By replacing shallow embeddings, e.g., Word2Vec, with context-aware LLM representations as the element features, i.e., $z_{i,k}$, SRAL can leverage a higher-quality semantic basis for its later learning.
> > - Furthermore, while LLMs are good at sequential modeling, they may face challenges with unordered and variable-sized set inputs. Therefore, it could be very promising to discuss their capability in this; and whether SRAL can function as a specialized structural adapter to compress complex set information into compact embeddings. These embeddings may then serve as soft prompts to guide LLMs in downstream reasoning tasks. We consider this as a promising direction for future research.
> >
> > We have discussed this in the revised Conclusion Section as follows:
> > > L.523: "Another direction is to explore the synergy with Large Language Models (LLMs), such as utilizing context-aware LLM representations as element features, or employing SRAL as a structural adapter to generate compact soft prompts for LLM-based set reasoning."
> >
> > **Q2**: In addition to $V\_O$, the learnable parameters $\Xi$ also include element embedding for $V_i$ of input set $S_i$. Furthermore, depending on the downstram main task, we also need to optimize neural network layers, e.g., ISAB for task 3, GNNs for task 2. All these parameters are jointly optimized via gradient descent to minimize the holistic objective $\mathcal{L} = \mathcal{L}\_{Main} + \lambda\_1 \mathcal{L}\_{Aux}$. Thank you.
> >
> > *We are deeply grateful to Reviewer kDXr for your detailed review and the time dedicated to improving our work. We have made every effort to address the concerns. We hope these revisions meet your expectations and warrant a reassessment of our work. Please let us know if there are any remaining aspects that need clarification.*

---

> ### Author Response · Authors · 2025-11-25
> **Follow-up on Rebuttal**
>
> Dear Reviewer kdXr,
>
> We sincerely appreciate the time and effort you dedicated to reviewing our work. We have posted a detailed response and a revised manuscript, in which we have made significant efforts to address all your concerns. Specifically:
>
> - We have substantially rewritten Sections 2 and 3 to simplify the presentation and reduce notation heaviness, as suggested.
>
> - We have added the requested comparisons to [1, 2, 3] on Tasks 1 and 4 to demonstrate the effectiveness of our method against recent approaches.
>
> - We have provided standalone results for Task 2 (without CrossCBR) and further clarified the significance of the performance gains.
>
> - We have enlarged the figures and tables to improve readability.
>
> Mindful of the potential Thanksgiving break and the rebuttal concluding in about one week (Dec 3rd), we wish to kindly reach out to ensure our response fully addresses all your questions. If there are any remaining issues, we remain fully available and active to provide further clarifications during the following time.
>
> Should you find that our revisions satisfactorily resolve all the issues, we wish to respectfully request a reassessment of our work and consideration of updating your evaluation scores.
>
> Thank you once again for your time, expertise, and constructive feedback that helped improve our paper.
>
> With sincere appreciation,
>
> Authors of Submission 2217

---

### Official Review · Reviewer_jqC5 · 2025-10-31

**Soundness:** 3
**Presentation:** 2
**Contribution:** 2
**Rating:** 6
**Confidence:** 4

**Summary:**

This paper presents an set representation learning approach that uses optimal transport to compare input sets and learnable references. The optimal transport is approximated via sliced Wasserstein distance (SWD) where the number of slices R is a hyperparameter (that trades between accuracy and computation time). In addition to the sliced Wasserstein distance used to encode sets (thus achieving permutation invariance and cardinality-independence), the authors also use adversarial-robust learning (in a min-max formulation) to be robust against the perturbations of encoded set features. The authors show that in a variety of tasks, using SWD to encode sets together with the adversarial-robust learning exceeds state of the art results.

**Strengths:**

This is a nice experimental paper, that extensively evaluates their proposed method and shows good results in all the tasks considered. The ablations also support the conclusion that both the OT-based encoding and adversarial-robust learning help to improve the performance in the tasks considered.

**Weaknesses:**

Unfortunately, OT in set-representation learning is not new, see Skianis et al. for what seems to be the first OT-approach. The authors do cite this paper in passing but not in detail. In that paper the OT approach is presented as a bipartite graph optimization, which is equivalent to the linear programming formulation for solving OT problems. The authors should discuss that paper in detail and compare their method against theirs (in a fair way, e.g. using the same encoders etc.) Interestingly, whereas the authors use the sliced Wasserstein distance to approximate the OT, the authors in that paper take another approach: by dropping one of the constraints, Skianis et al. reduces OT to the semi-relaxed case (unbalanced OT with one of the penalties going to infinity). The authors should also extend their comparisons to Skianis et al. by comparing the various approximations.

**Questions:**

- I found the intro to be hard to crack, given that I have a background in OT and deep learning but not set representation learning (SRL). I had to look into some of the author's references to understand the topic better. Perhaps the problem description in Preliminaries could be extended with a figure or a motivational example could be given to introduce SRL to a wider audience.
- how do you define a CDF in d-dimensions (line 159) ? You can only do this after projecting into 1-d (unless you introduce 'copulas'!)
- not clear from intro how the 'reference distribution' O is learned!
- For Proposition 1 it is enough to cite a standard book on OT, e.g. Peyre and Cuturi's book. As it is a well known result, no need to cite it as a Proposition, because it's not the author's contribution.
- Similarly for Proposition 2, sorting and matching the quantiles is well known in the literature as a solution, no need to state it as a proposition.
- It is not clear how the features z_{i,k} of the sets are obtained from the intro.
- Besides Skianis et al., the papers by Mialon et al. (2021) [OTKE] and Kim (2022) both discuss OT-approaches in a kernel-method context. These and other OT-based methods should be discussed in detail and compared against.
- Are all datasets in the experiments supervised? Discussing the experiments a bit better in the main text would be better.
- Figure 3: A and B captions should be swapped
- very hard to read most of the figures, please revise
- not clear what is the difference between AEPO and AL in the ablation study.

---

> ### Author Response · Authors · 2025-11-21
> **Response to Reviewer jqC5 (Part I)**
>
> We sincerely appreciate Reviewer jqC5 for your professional and constructive feedback. We have addressed each of your comments and suggestions below and highlighted the corresponding revisions in blue in the updated manuscript.
>
> **W1**:
> - We thank Reviewer jqC5 for this constructive suggestion. We wish to clarify that our contribution extends beyond merely introducing OT to set representation learning and further lies in the proposed Adversarial Encoding Perturbation and Optimization (AEPO) mechanism. This framework constructs a robust auxiliary learning objective, enabling the model to learn set representations compatible with diverse downstream tasks.
> - To address the comparison with [Skianis et al., 2020] in detail, we have expanded the Related Work section with the following discussion:
>     > L.834: "Additionally, RepSet is one of the earliest approaches to explicitly incorporate Optimal Transport into set representation learning. To capture complex set structures, RepSet generates embeddings by solving a Bipartite Matching problem that optimally aligns the elements of an input set with a collection of learnable hidden sets. However, unlike this high-dimensional formulation, our SRAL employs the 2-Sliced-Wasserstein metric to bypass the computational bottleneck of exact matching via effective approximation."
>
>     Furthermore, following your suggestion, we conducted an extensive comparative analysis covering three aspects: (1) SRAL vs. RepSet (baseline performance), (2) the effectiveness of RepSet as an encoder within our framework, and (3) the performance of the approximation variant, ApproxRepSet. Due to time limit, we report the preliminary results on Task 1 (Friendster) below and will update results of the remaining tasks later.
>
>     |  |SRAL|RepSet|SRAL (RepSet)|ApproxRepSet|SRAL (ApproxRepSet)|
>     |-----|-----|-----|-----|-----|-----|
>     |Recall@20| 91.57     |  80.63 |82.25  |77.81 | 78.93|
>
>     (1) We observe that vanilla RepSet performs comparably to other baselines but underperforms SRAL. While integrating RepSet with our AEPO framework yields a slight performance gain, this validates our analysis in Remark 1: our AEPO mechanism is specifically tailored for the Set Feature Encoder (SFE), where adversarial perturbations are provably aligned with the optimization of the Sliced-Wasserstein metric. This however may not be aligned within the RepSet method. (2) As for the approximation method in [Skianis et al., 2020], i.e., ApproxRepSet, we observe a similar trend. We attribute its inferior performance to the nature of its "semi-relaxed" approximation. According to its original paper, after relaxing constraints, ApproxRepSet may sacrifice rigorous metric properties (e.g., the triangle inequality) and is prone to local optima. In contrast, the Sliced-Wasserstein approach preserves key geometric properties, resulting in superior overall performance. Due to page limit, we have currently updated these empirical results in Appendix E.10.

---

> > ### Author Response · Authors · 2025-11-21
> > **Response to Reviewer jqC5 (Part II)**
> >
> > **Q1**:
> > Thank you Reviewer jqC5 for the constructive suggestion.
> > In the Intro section, we have specifically added a description of Set Representation Learning as follows:
> > >L.47: "To address this limitation, Set Representation Learning has emerged. Unlike naive approaches that merely sum element features, its fundamental goal is to learn a holistic and fixed-size embedding that captures the intrinsic semantics of the whole unordered collection. This capability is essential for facilitating complex downstream tasks, for example, in E-commerce, enabling the model to interpret a product bundle as a cohesive semantic unit, e.g., a "camping kit", rather than a loose collection of items, thereby improving recommendation accuracy."
> >
> > Furthermore, we have also added a more detailed illustrative example after the Problem Description of preliminary Section:
> > > L.112: "Illustrative Example. Take bundle recommendation as a concrete example. A set $S_i$ represents a product bundle, e.g., a "camping kit", containing items such as a tent, a sleeping bag, a kettle, etc. The necessity of Set Representation Learning lies in its ability to compress these irregular structures into a unified and fixed-size embedding. This enables downstream systems to interpret the bundle as a coherent semantic entity that captures the collective information of the items, rather than treating them as a loose collection of isolated products. Technically, the learned set representations should accommodate varying cardinalities, i.e., different numbers of items across bundles, and ensure permutation invariance, meaning the resulting embedding remains identical regardless of the order in which the items are listed."
> >
> > We hope these revisions effectively enhance the manuscript readability and fully address your concerns.
> >
> >
> > **Q2**:
> > Thank you Reviewer jqC5 for the feedback. Our intention was to define an empirical **`Multivariate CDF`** in $\mathbb{R}^d$ space. We have clarified this in the revision as follows:
> > > L.177: "Therefore, we can define the empirical Multivariate Cumulative Distribution Function (CDF) of $P_i$ as: $F\_{{P}\_i}(\boldsymbol{x}) = \frac{1}{|S\_i|} {\sum}\_{k=1}^{|S\_i|} \delta(\boldsymbol{x} \geq \boldsymbol{z}\_{{i, k}})$, where $\delta(\cdot)$ is the indicator function$^1$. "
> >
> > and
> >
> > >"$^1$The comparison $\boldsymbol{x} \geq \boldsymbol{z}\_{{i, k}}$ is performed element-wise. That is, $\delta(\cdot)$ returns 1 if and only if $\boldsymbol{x}\_j \geq \boldsymbol{z}\_{{i, k, j}}$ for all dimensions $j=1, \ldots, d$, and 0 otherwise."
> >
> > Please let us know if further clarification is needed.
> >
> > **Q3**:
> > We appreciate Reviewer jqC5 for pointing this out. We clarify that $O$ is characterized by $H$ $d$-dimensional trainable embeddings, i.e., $[\boldsymbol{z}\_{h} \in \mathbb{R}^d]\_{h=1}^{H}$.
> > Then we do not learn a distribution function explicitly; instead, we learn the $H$ embedding vectors that constitute $O$. They are optimized end-to-end via standard gradient descent alongside other network parameters to minimize the task loss. We have added a specific description in Section 3.2 to clarify this learning mechanism.
> > > L.181: "This reference, characterized by $H$ trainable embeddings, .... Specifically, these embeddings are initialized as model parameters and are updated via backpropagation during the training process, adapting globally to minimize the overall objective function."
> >
> > **Q4**:
> > Thank you Reviewer jqC5 for the suggestion. Previously, we cited previous work and included the formal proposition to ensure self-containment and readability.
> > Following your suggestion, we have now referenced the standard literature in the main text and moved the formal statement to Appendix D.
> >
> > **Q5**: Thanks Reviewer jqC5 for the advice. We have already done so in revision.
> >
> > **Q6**: $\boldsymbol{z}_{i,k}$ represents the $d$-dimensional embedding vector associated with the $k$-th element of set $S_i$. In practice, the initialization of these embeddings depends on the availability of raw features. In revision, we make further explanation in Problem Description as:
> > > L.106: "Each element $e_j \in \mathcal{E}$ is associated with a $d$-dimensional embedding vector $\boldsymbol{z}_j \in \\{\boldsymbol{z}_1, \boldsymbol{z}_2, \cdots, \boldsymbol{z}_n\\}$. In practice, depending on the availability of raw features, these embeddings $\boldsymbol{z}_j$ are either initialized using pre-trained feature extractors, e.g., word embeddings, or randomly initialized to be learned from scratch."

---

> ### Author Response · Authors · 2025-11-21
> **Response to Reviewer jqC5 (Part III)**
>
> **Q7**: We appreciate Reviewer jqC5 for your constructive suggestions. Due to time limits, we have updated results on Task 1 and Task 4 as follows. Results of remaining tasks will be updated later.
> Based on the following results, we notice that, both OTKE [Mialon et al. (2021)] and DIEM [Kim (2022)] still underperforms our model. This is likely because they prioritize intra-set dependencies, e.g., self-attention in OTKE, while may not explicitly capture inter-set similarities. Additionally, the absence of an enhanced training approach like our AEPO may further limit the effectiveness of their set representations.
>
> | Task 1 | R@20 (FS) | N@20 (FS) | R@100 (FS) | N@100 (FS) | R@20 (LJ) | N@20 (LJ) | R@100 (LJ) | N@100 (LJ) |
> | :--- | :--- | :--- | :--- | :--- | :--- | :--- | :--- | :--- |
> | OTKE | 79.53 | 73.68 | 86.64 | 79.59 | 81.45 | 79.82 | 87.95 | 85.10 |
> | DIEM | 82.49 | 81.40 | 88.36 | 81.56 | 83.95 | 84.92 | 89.88 | 87.15 |
> | SRAL | 91.57 | 92.22 | 94.53 | 93.01 | 87.56 | 89.31 | 92.93 | 91.25 |
>
>
> | Task 4 | LDA-1k |LDA-3k |LDA-5k |
> | :--- | :--- | :--- | :--- |
> | OTKE | 62.95| 77.59| 72.57|
> | DIEM | 63.58|75.67 | 76.96|
> | SRAL | 80.94 | 87.93| 86.20|
>
> **Q8**:
> Yes, all datasets used in our experiments are supervised.
> While our proposed SRAL framework introduces a self-supervised auxiliary objective, it is designed to work jointly with a supervised main task objective ($\mathcal{L}_{Main}$). We make this clarification in revision. Thank you Reviewer jqC5 for the suggestion.
> > L.320: "All tasks and datasets conducted in our experiments are in a supervised setting."
>
> **Q9,Q10**: Thank you Reviewer jqC5 for the reminder. We have fixed the first issue. We have also addressed the readability concern, by increasing the font size and optimizing the layout of Figs. 1,3,4 to ensure they are clear and legible. Please let us know if there are any remaining issues requiring further adjustment.
>
> **Q11**: We are sorry for the confusion. We clarify the distinction between these two variants as follows:
> (1) Variant `w/o AEPO` keeps the auxiliary learning objective but disables the adversarial optimization step. Specifically, we utilize the inner InfoNCE loss but remove the min-max strategy, meaning the model is trained without generating worst-case perturbations via gradient ascent. (2) Variant `w/o` removes the entire auxiliary learning framework and the model is trained solely on the main task supervision. We have revised the descriptions in Section 4.3.3 to explicitly articulate them and eliminate any ambiguity:
> > L.501: "3. Variant "w/o AEPO" retains the auxiliary learning objective but disables the adversarial optimization step. Specifically, we utilize the inner InfoNCE loss but remove the min-max strategy that  generates worst-case perturbations via gradient ascent. The results show that it negatively impacts performance, particularly on Task 4 where the AUC drops by 24.70%. 4. In contrast, removing the entire auxiliary learning ("w/o AL") by training the model solely with the main task supervision, results in a marginal performance improvement for Task 2."
> > L.320: "All tasks and datasets conducted in our experiments are in a supervised setting."
>
>
> *We hope our responses effectively address Reviewer jqC5's concerns. We appreciate your constructive feedback and remain available for any further clarification.*

---

> > ### Author Response · Authors · 2025-11-24
> > **Response to Reviewer jqC5 (Part IV)**
> >
> > Dear Reviewer jqC5:
> >
> > We have supplemented the experiments for Task 3. We observe that these two methods perform comparably to the other methods discussed in the paper. Additionally, we have updated all experimental results and discussions in the revised paper. Thank you.
> >
> > | Task 3 | OTKE | DIEM | SRAL |
> > | :--- | :--- | :--- | :--- |
> > | MLP | 85.92 | 85.58 | 86.53 |
> > | ISAB | 86.70 | 86.72 | 87.31 |

---

> > > ### Comment · Reviewer_jqC5 · 2025-11-27
> > > **some minor comments and questions**
> > >
> > > Thank you for your carefully prepared rebuttal. I made some new notes while going through your revised manuscript:
> > >
> > > - The correction in line 126 is wrong: x and g(x) cannot be compared
> > >   if g outputs are one dimensional!
> > > - The closed form solution given in line 133 is not correct: it should
> > >   be the other way around (see e.g., 2.39 in Peyre et al.)
> > >   (the solution given in 196 is correct)
> > > - you don't need to define a multivariate CDF (which uses the unintuitive
> > >   elementwise comparison for sorting vectors) since you only work in the
> > >   sliced setting.
> > > - Besides the illustrative example given, it would be nice to add
> > >   some references where more examples can be found.
> > > - Make sure to proofread your text carefully, there are various typos.
> > > - Can't eq.5 be reduced to the second case (subsumes the first!)
> > > - In Remark 1, shouldn't the expectation be on the left-hand side?
> > >   It seems weird the way it is formulated now.
> > > - objectives -> parameters in line 275
> > > - W1 of reviewer du3f: this is an interesting question. It seems
> > >   other OT based set representation methods also fix learnable references
> > >   to compare distributions, this could be mentioned in the main text.
> > >   Could you instead average the different embeddings computed
> > >   between each set pair in a batch to prevent the highly unstable
> > >   training?
> > > - "our AEPO mechanism is specifically tailored for the Set Feature
> > >   Encoder (SFE), where adversarial perturbations are provably aligned
> > >   with the optimization of the Sliced-Wasserstein metric."
> > >   Indeed, if the authors infer from the experiments this conclusion,
> > >   then this fact should be discussed in the introduction and the
> > >   main text and emphasized. As of now, I am unable to understand/appreciate
> > >   this 'provable alignment' from the text.

---

> > > > ### Author Response · Authors · 2025-11-27
> > > > **Response to Reviewer jq5C's Latest Comments (Part I)**
> > > >
> > > > Dear Reviewer jqC5,
> > > >
> > > > We sincerely appreciate Reviewer jqC5 for the detailed follow-up discussions and constructive suggestions. We are deeply grateful for your continued engagement to improve our work. Our response to your specific feedback is as follows:
> > > >
> > > > - Q1: We thank Reviewer jqC5 for pointing out this inconsistency. We acknowledge that this confusion arose from overloaded notation. In Eq. (1) and the subsequent theoretical context, $g$ and $g^+$ denote the transport plan and the optimal transport map between distributions $P$ and $Q$, which operate within the same dimensionality. However, in L.130, the symbol $g$ was inadvertently used to denote a projection function, creating ambiguity. To resolve this, we have renamed the function in line 126 to $\theta$. This correction ensures consistency with the projection notation used throughout the rest of the paper (e.g., as defined in Eq. (2) and Section 3.2). We apologize for the confusion and have updated the manuscript accordingly.
> > > >     > L. 130: For the projection function $\theta: \mathbb{R}^d \rightarrow \mathbb{R}$, $P^{\theta}$ represents the push-forward of ${P}$ with $\theta$ in a one-dimensional space, defined as $P^{\theta}({Y}) = P({{x}: \theta({x}) \in {Y}}) = P(\theta^{-1}({Y}))$.
> > > >
> > > > - Q2: We wish to respectfully clarify that in Peyré et al. (2019), the formulation $T=\mathcal{C}^{-1}\_\beta \circ \mathcal{C}\_\alpha$ (Eq. 2.39) represents a composition of functions that operates from right to left. Specifically, it first computes the CDF of the source distribution $\alpha$, followed by the inverse CDF of the target distribution $\beta$. This defines the optimal transport map from source $\alpha$ to target $\beta$. Thus, this formulation is mathematically equivalent to the content in L.138. However, we acknowledge that L.138 requires an explicit specification regarding the direction of transport, which is from $Q$ to $P$ in our notation. As the reviewer correctly noted, while the Wasserstein metric itself is symmetric, the implementation of the transport map is directional. Please kindly notice that, we adopted this specific formulation to ensure consistency with our subsequent methodology, where the reference distribution $O$ serves as the source for the transport map (as detailed in Section 3.2 2). Then to prevent ambiguity, we have added an explanatory note at Line 138 to explicitly state the transport direction. We hope this clarification resolves the concern.
> > > >     >L.138: "If a minimizer exists, denoted by ${g}^+$, it is the solution to the OT problem. For one-dimensional distributions, a closed-form tranport solution for ${g}^+$ from $Q$ to $P$ exists: ..."
> > > >
> > > > - Q3: We appreciate the reviewer's constructive suggestion. We agree that the preliminary discussion of the Multivariate CDF is not strictly necessary for the subsequent analysis. Our original intention was to present the general form first, allowing the one-dimensional CDF in the sliced setting to be treated as a special case for conciseness. However, we recognize that simplifying this exposition can improve the paper's readability. Therefore, we have removed the multivariate definition from the preliminaries and now introduce the CDF directly in the context of Eq.(4), as follows:
> > > >     > L. 202: "Here the CDF of $O^\theta$ is: $F_{{O}^\theta}({x}) = \frac{1}{H} {\sum}_{h=1}^{H} \delta(x \geq \boldsymbol{w}^\mathsf{T}\boldsymbol{z}_h)$, where $\delta$ returns 1 for zero input and 0 otherwise."
> > > >
> > > > - Q4: Thanks for the suggestion. We have added several other concrete examples as:
> > > >     > L.120: "Beyond the specific e-commerce scenario illustrated above, Set Representation Learning has found broad applications across diverse domains. These include computer vision, for aggregating multi-view images or video frames [wang2022exploring], bioinformatics, where proteins are modeled as sets of residues for property prediction [naderializadeh2025aggregating], and computational pathology, where whole-slide images are treated as sets of patches for cancer diagnosis [ilse2018attention, carbonneau2018multiple]."
> > > >
> > > > - Q5, Q6. Thank you Reviewer jqC5 so much for the advice. We will continue checking the paper and simplify Eq.(5) as suggested.
> > > >
> > > > - Q7: We appreciate this helpful suggestion. Our original intention was to present the left-hand side as a Monte Carlo approximation of the ground-truth value on the right. However, we have followed your advice and moved the expectation operator to the left-hand side. This revision more accurately formalizes the embedding distance as an estimator of the true distance. We have updated the equation in revision accordingly.
> > > >
> > > > - Q8: Has modified. Thx.

---

> > > > > ### Author Response · Authors · 2025-11-27
> > > > > **Response to Reviewer jq5C's Latest Comments (Part II)**
> > > > >
> > > > > - Q9: We have supplemented the manuscript with discussions on related works that share similar design concepts, as follows:
> > > > >     > L.184: "This reference design, characterized by $H$ trainable embeddings ${V}\_O = [{z}\_{h} \in \mathbb{R}^d]\_{h=1}^{H}$, as a strategy shared by several previous methods [guo2021learning,mialon2021trainable,naderializadeh2021pooling], serves as a learnable ``origin'' in the set embedding space."
> > > > >
> > > > >
> > > > >
> > > > >     Regarding the second inspiring discussion: Inspired by Reviewer jqC5, we explored a batch-based averaging strategy. Specifically, for a target set $S_i$, we computed its representation by averaging the embeddings derived from interactions with all other sets in the current batch (i.e., averaging over $\{ (S_i, S_a), (S_i, S_b), \dots \}$). This approach actually incorporates the idea of *Ensembling*, which helps to mitigate noise through information aggregation. We conducted a rapid verification on Task 4 as follows, and the results indicate that this variant yields inferior performance compared to our original design. We preliminarily attribute this performance gap to two factors: (1) this approach may fail to preserve the geometric properties of the SW distance. As this may thus introduce distance distortion in the embedding space, the model's ability to capture precise distributional correlations is hindered. (2) Within each batch, the specific pairs associated with $S_i$ are random and non-fixed. Consequently, the number of samples available for averaging is unstable across iterations. This inconsistency in the aggregation scope may introduce additional variance, which likely further compromises the effectiveness of Ensembling mechanism.
> > > > >
> > > > >
> > > > >     |AUC|SRAL (origin)|SRAL (ensemble)|
> > > > >     |---|---|---|
> > > > >     |LDA-1K|80.94|72.35|
> > > > >     |LDA-3K|87.93|77.58|
> > > > >     |LDA-5K|86.20|77.82|
> > > > >
> > > > >
> > > > > - Q10: We appreciate Reviewer jqC5's rigorous suggestion. We acknowledge that the phrase "provably aligned" may have been imprecise and potentially confusing. Our original intention was to convey that, as demonstrated in Remark 1, the perturbations and optimization objectives within our AEPO design are positively correlated in expectation with the underlying SW metric. This consistency supports our empirical observation that AEPO effectively enhances the quality of set embeddings. To ensure clarity and avoid ambiguity, we have revised the manuscript to use the more precise term "correlated," consistent with our earlier derivation.
> > > > >     > L.1312: "The adversarial perturbations generated by AEPO aim to maximize discrepancies in the embedding space, which are positively correlated in expectation with the Sliced-Wasserstein distance. In contrast, such a direct perturbation may not hold for the bipartite matching objective in RepSet, rendering the adversarial optimization less effective in capturing distributional semantics."
> > > > >
> > > > >
> > > > > *We sincerely appreciate Reviewer jqC5's recognition of our work, particularly your decision to maintain a positive score and your expressed hope for the paper's acceptance. We hope that our above responses have fully addressed your concerns. Should any ambiguity remain or if further clarification is needed, please do not hesitate to let us know.*
> > > > >
> > > > > *Lastly, please pardon our forwardness: given the highly competitive nature of ICLR, if the reviewer finds that our revisions have strengthened the quality and rigor of the manuscript, we would be deeply grateful if you might consider raising the score. This would significantly support the final evaluation of our work. Regardless, we fully respect your professional judgment and final decision. Thank you once again for your time and your dedication to the ICLR review.*
> > > > >
> > > > > With sincere appreciation,
> > > > >
> > > > > Authors of Submission 2217

---

> ### Author Response · Authors · 2025-11-25
> **Follow-up on Rebuttal**
>
> Dear Reviewer jqC5,
>
> We sincerely appreciate the time and effort you dedicated to reviewing our work. We are sorry for the delay taken to complete all the experiments and revisions. We have now posted a detailed response and revised the manuscript.
>
> Specifically, we have conducted the requested comparative experiments with RepSet and discussed other OT-based methods in detail. We have also significantly revised the Introduction and Preliminaries to improve clarity, clarified the CDF definition, adjusted Propositions 1 & 2, and improved the figure readability, etc.
>
> Aware of the potential Thanksgiving break and the discussion period concluding in about one week (Dec 3rd), we wish to kindly reach out to ensure our response fully addresses your concerns. If there are any remaining questions, we will be ready to provide explanations at the following time. Should you find that these new clarifications and revisions have fully resolved your concerns, we would be most grateful if you could consider updating your evaluation score accordingly.
>
> Thank you once again for your time, expertise, and support.
>
>
> With sincere appreciation,
> Authors of Submission 2217

---

### Official Review · Reviewer_9yXr · 2025-10-31

**Soundness:** 3
**Presentation:** 3
**Contribution:** 2
**Rating:** 6
**Confidence:** 4

**Summary:**

This paper proposes an optimal transport-based approach for learning cardinality- and permutation-invariant representations for set-structured data. A set encoder based on sliced Wasserstein distance is used, and a contrastive-style adversarial loss is added to the objective to learn more informative set embeddings based on worst-case perturbations of the element-wise embeddings. Numerical results on four different tasks show the superiority of the proposed approach over several other set learning methods.

**Strengths:**

- Learning representations of sets is a critical problem in many application areas of machine learning, so the considered problem is timely and important.
- The paper is written clearly and relatively straightforward to read and follow, even for an audience with minimal background in set representation learning.
- The scope of the presented experiments is broad and acceptable in my view.

**Weaknesses:**

- It is unclear how the optimal adversarial perturbations are applied to the sets. On line 249, the authors mention that the perturbations are shared between all sets and both views of each set. However, the two views of each set will be identical if the perturbations are equal. Is one view unperturbed and the other view perturbed by $\epsilon^+$? Is $\epsilon^+$ the perturbation used for all sets in a given training batch?
- In the abstract and introduction section, the authors mention that perturbations are not applied to the input features but occur during the encoding process. However, as presented, it does seem that the perturbations are indeed applied to the features *before* the encoding process. If my understanding is correct, the claims in the abstract and introduction need to be revised. Otherwise, if the encoding process itself is to be perturbed, one option is to perturb the projection parameters $\Theta=[w_r]_{r=1}^R$; would that be feasible with your adversarial approach?
- If I understand correctly, the SFE module introduced in Section 3.2.1 is now new and is the same as the one introduced by PSWE [A]. Could you please explain if there are any distinctions between SFE and PSWE?
- The notation at the beginning of Section 2 is confusing. Is the universe $\mathcal{E}$ finite and contains $n$ elements, or is $n$ the cardinality of the set $S_i$? What is the difference between $e_i$'s and $z_i$'s?
- The contribution of Remark 1 is unclear to me. SFE and PSWE both imply that the Euclidean distance between embeddings approximates the Sliced Wasserstein distance before the encoding process, which is unrelated to the used InfoNCE loss function. Also, could you please explain more precisely how you go from Eq. (8) to Eq. (9) (from summation to integration)? Do you need assumptions on, e.g., an infinite number of projections $R$?

[A] Navid Naderializadeh, Joseph F Comer, Reed Andrews, Heiko Hoffmann, and Soheil Kolouri. Pooling by sliced-wasserstein embedding. NeurIPS, 34:3389–3400, 2021.

**Questions:**

- Is it possible to simplify Eq. (13) by setting $\epsilon^+ = \frac{\pi}{\\|g_{\epsilon^+}\\|} g_{\epsilon^+}$?
- Could you please compare your method's performance with some more recent approaches, such as FSW [B]?

[B] Amir, Tal, and Nadav Dym. "Fourier Sliced-Wasserstein Embedding for Multisets and Measures." In The Thirteenth International Conference on Learning Representations, 2025.

---

> ### Author Response · Authors · 2025-11-21
> **Response to Reviewer 9yXr (Part I)**
>
> We sincerely appreciate Reviewer 9yXr for your insightful review and suggestions. We provide detailed responses to each of the raised concerns as follows, hoping to clarify these points satisfactorily. In revision, we specifically use blue to highlight corresponding modifications.
>
> **W1**:
> We are sorry for the confusion caused. We intended to convey that two perturbed views, $V_i'$ and $V_i''$, are first constructed via Eq. (7), where $V_i'=[\boldsymbol{z}'\_{i,k}]_{k=1}^{S_i}$ and $V_i''=[\boldsymbol{z}''\_{i,k}]\_{k=1}^{S_i}$. It is important to note that the random perturbations within $V_i'$ and $V_i''$ are independent and non-identical. Then the mentioned term $\epsilon^+$ (*in revision, we modify it as $\sigma \to \epsilon^+$ for better readability*) represents a small increment that is subsequently superimposed onto these views. Therefore, the resulting views remain non-identical. To eliminate any ambiguity, we have revised the text in Line 253 of the updated manuscript as follows:
> > L.273: "We seek an adversarial perturbation increment $\boldsymbol\sigma$. $\boldsymbol\sigma$ is shared and applied to the perturbed features ${V}'_i$ and ${V}''_i$ that are generated earlier from Eq. (7)."
>
> **W2**:
> (1) We are very grateful for Reviewer 9yXr's valuable feedback. We acknowledge that our description in the Abstract and Intro. was not sufficiently precise and led to ambiguity.
> - Our original intention was to highlight the fundamental difference between our method and traditional **`data-level`** augmentations (e.g., element dropout/addition or subset sampling). Our method however operates at the **`feature-level`**.
> - The reviewer's observation is accurate: our adversarial perturbations are indeed applied to the feature embeddings before they are processed by the SFE. We used the phrase "perturbs the set encoding process itself" mainly to emphasize a critical distinction: they are not a pre-defined pre-processing step (like adding random noise only); instead, as detailed in Section 3.3.2, these perturbations are endogenously and adversarially generated and "learned" from our min-max optimization objective (Eq. 10). Its calculation depends on the gradient of the auxiliary loss $\mathcal{L}_{wd}$ (Eq. 8) with respect to the perturbation itself. Therefore, the generation of this perturbation is tightly coupled with the encoding and optimization framework, rather than being a simple manipulation of the input.
> - To eliminate this ambiguity, we have revised  the abstract (in the updated version) as follows:
>     > L.22: "Instead of manipulating the input data, e.g., element dropout/addition, our method introduces adversarial perturbations at the feature level. Through min-max optimization, we compel the model to achieve robustness against worst-case perturbations."
>
>     and Intro.:
>     > L.88: "Specifically, our approach departs from conventional data manipulation strategies, e.g., element dropout/addition or subset sampling, by introducing adversarial perturbations directly to the set features."
>
>     as well as other modifications, e.g., L.160, L.519. We hope they can improve the clarity of our paper.
>
> (2) As for the second question, however, perturbing the projection parameters $\Theta$ may be inconsistent with our theoretical framework. As defined in Section 2, Sliced-Wasserstein requires the projection $\theta(x) = w^\top x$ to use a unit vector $w$ that is uniformly sampled. Then simply adding the perturbation to $w$ may violate the requirements. Therefore, to maintain theoretical consistency, in this work, we apply the adversarial perturbation mainly to the set features.
>
> **W3**:
> Thank you Reviewer 9yXr for this insightful question. Our SFE module shares its foundational principles with pswe, as both methods are grounded in the Sliced-Wasserstein distance to "discretize" the high-dimensional represenations. The core similarity lies in the feature projection into 1D space and straightforward implementation of the optimal transport map $g^+$. However, a key distinction lies in the final aggregation step used to construct the set embedding. While PSWE employs a weighted sum to aggregate the transport results from the different slices, our SFE module, as defined in Eq. (6), uses direct concatenation. We opted for concatenation to preserve the distance information with less compressed and better approximation of Sliced-Wasserstein distance. In addition, please kindly notice that for a fair comparison, we kept the same set embedding size for them in our experiments.

---

> ### Author Response · Authors · 2025-11-21
> **Response to Reviewer 9yXr (Part II)**
>
> **w4**:
> Thank you Reviewer 9yXr for pointing out this ambiguity. To fist answer the reviewer's questions directly: $\mathcal{E}=\\{e_{1},e_{2},\cdot\cdot\cdot,e_{n}\\}$ is indeed the finite universe containing $n$ unique elements. $\boldsymbol{z}_j \in \mathbb{R}^d$ refers to the $d$-dimensional embedding vector corresponding to the element $e_j$. We have revised the "Problem Description" as follows and would appreciate your feedback on whether any further clarification is needed:
> > L. 105: "Let $\mathcal{S}=\\{S_{1},S_{2},\cdot\cdot\cdot,S_{m}\\}$ be a corpus of $m$ sets.The elements in these sets are drawn from a finite universe $\mathcal{E} = \\{e_1, e_2, \cdot\cdot\cdot, e_n\\}$, which contains $n$ unique elements. Each element $e_j \in \mathcal{E}$ is associated with a $d$-dimensional embedding vector $\boldsymbol{z}_j \in \\{\boldsymbol{z}_1, \boldsymbol{z}_2, \cdots, \boldsymbol{z}_n\\}$. "

---

> ### Author Response · Authors · 2025-11-21
> **Response to Reviewer 9yXr (Part III)**
>
> **w5**:
> (1) We are grateful for Reviewer 9yXr's valuable feedback.
> - The core premise of our SFE module is consistent with that of PSWE: the encoder is designed so that its output embedding conforms to the Sliced-Wasserstein (SW) distance measurement.
> - However, our work (Section 3.3) introduces an adversarial perturbation and optimization scheme on top of this. This raises a critical question: does this new auxiliary task $\mathcal{L}_{wd}$ disrupt or interfere with the SFE module's original capability to approximate the SW distance? The contribution of Remark 1 is thus to fill this gap. It serves as a bridge to demonstrate that our InfoNCE-based self-supervised objective is, in expectation, equivalent to optimizing the objective directly on the underlying SW distances between the perturbed high-dimensional distributions. This proves that our goal enables the model to learn robust representations within the SW distance space, thereby reinforcing, rather than disrupting, the set representation learning capability of our approach.
> To clarify this, we have revised the text to:
>     > L.255: "This however raises a question: does this learning objective, which operates on perturbed embeddings, disrupt the SFE module's fundamental capability derived from the Sliced-Wasserstein metric? Therefore, we introduce Remark 1, which demonstrates that optimizing this objective is, in expectation, equivalent to optimizing the objective directly on the underlying distributional distances, thus consolidating the learning capability of our set embedding approach."
>
> (2) We thank the reviewer for raising this precise question. The transition from Eq. (8) to Eq. (9) indeed requires $R \to \infty$. Our previous brief reference to the "continuous form in expectation" was originally based on the assumption of "sufficient Monte Carlo approximation". We have clarified this in the revised manuscript to ensure rigor. As for the process from Eq. (8) to Eq. (9), from Proposition 1, we know that ${g}^+(x^{\theta}|{V}\_i^{\theta}) = F^{-1}\_{{P_i}^{\theta}}\big(F\_{{O}^{\theta}}(x^{\theta})\big)$ (*please notice that, in revision, we simplify the notations from ${P\_i}\_{<\theta>}$ to ${P\_i}^{\theta}$, and ${O}\_{<\theta>}$ to ${O}^{\theta}$*). Then the RHS of Eq.(8) becomes:
>
> $$\begin{aligned}
> & \Big\| \frac{1}{R} {\sum}\_{r=1}^{R} \frac{1}{H} {\sum}\_{h=1}^{H} \left({g^{+}}(
> {w}\_r^{\mathsf{T}}{z}\_{h}|{V}\_i'^{{\theta}\_r}) - {g^{+}}({w}\_r^{\mathsf{T}}{z}\_{h}|{V}\_j''^{{\theta}\_r}) \right)^2 \Big\|_2 \\\\
> &=\Big\| \frac{1}{R} {\sum}\_{r=1}^{R} \frac{1}{H} {\sum}\_{h=1}^{H} \left(F^{-1}\_{{P'_i}^{\theta\_r}}\big(F\_{O^{\theta\_r}}({w}\_r^{\mathsf{T}}{z}\_{h})\big) - F^{-1}\_{{P''\_j}^{\theta\_r}}\big(F\_{O^{\theta\_r}}({w}\_r^{\mathsf{T}}{z}\_{h})\big) \right)^2 \Big\\|_2
> \end{aligned}$$
>
> The inner summation is substituted by recognizing the relationship between the sample mean and the integral for an empirical distribution. Given an empirical distribution $P(x)=\frac{1}{H}\sum_{h=1}^H\delta(x\geq X_h)$, the sample mean equals the integral: $\frac{1}{H}\sum_{h=1}^H f(X_h) = \int f(x)dP(x)$.Then we apply this to the inner summation, with $t={w}\_r^{\mathsf{T}}{z}\_{h}$ being a sample from $O^{\theta\_r}$:
> $$\begin{aligned}
> & \frac{1}{H} {\sum}\_{h=1}^{H} \left(F^{-1}\_{{P'\_i}^{\theta\_r}}\big(F\_{O^{\theta\_r}}(t)\big) - F^{-1}\_{{P''\_j}^{\theta\_r}}\big(F\_{O^{\theta\_r}}(t)\big) \right)^2\\\\
> = & \int \left(F^{-1}\_{{P'\_i}^{\theta\_r}}\big(F\_{O^{\theta\_r}}(t)\big) - F^{-1}\_{{P''\_j}^{\theta\_r}}\big(F\_{O^{\theta\_r}}(t)\big) \right)^2 dO^{\theta\_r}(t)
> \end{aligned}$$
> For the outer summation, as $R \to \infty$, we apply the Law of Large Numbers for integration over the hypersphere: $\frac{1}{R} \sum\_{r=1}^{R} f(\theta\_r) = \int_{\mathbb{S}^{d-1}} f(\theta) d\theta$. Then integrating the result completes the derivation from Eq. (8) to Eq. (9).
>
> We have extended the above process into our revision (Appendix D). We also noted a minor oversight: $||v_i'-v_j''||_2$ is proportional ($\propto$) to the derived $RHS$ (scaled by $\sqrt{R \cdot H}$). This does not affect the conclusion of Remark 1, as the constant coefficient is canceled out by the InfoNCE loss structure. Please let us know if there is any part that remains unclear. Thank you.

---

> ### Author Response · Authors · 2025-11-21
> **Response to Reviewer 9yXr (Part IV)**
>
> **Q1**:
> - Thank you for the professional suggestion. The proposed simplified form, $\sigma=\frac{\pi}{||g\_{\sigma}||\_2}g\_{\sigma}$ (*please notice that, in revision, we simplify the notation from $\epsilon^+$ to $\sigma$*), is the closed-form solution for the inner maximization problem $max\_{||\sigma||\_{2}\le\pi}\mathcal{L}\_{wd}(\Xi,\sigma)$ when derived from the first-order Taylor approximationn. As we also discussed in Appendix D, Eq.(10), this form represnts the optimal perturbation.
> - However, in our practical implementation, as detailed in Section 3.3.2, we employ an alternating two-step approach to optimize this inner maximization problem. The reason is mainly to stabilize the training and to prevent overly aggressive perturbations. The exact solution always forces the perturbation's magnitude $||\sigma||_2$ to equal the maximum constraint $\pi$. During the initial phases of training, when model gradients are unstable, such maximal perturbation can lead to overly aggressive parameter updates and potentially cause training divergence. Therefore, we introduce the ascent step size $\eta$ to limit the norm of the perturbation $\hat{\sigma}$ less than $\pi$. In this case, Eq.(13) returns $\sigma = \hat{\sigma}$, but not the above exact solution. This method actually allows us to regularize the model with a milder and more controlled perturbation magnitude, which demonstrates to enhance the training robustness and convergence.
>
> **Q2**:
> We sincerely appreciate Reviewer  9yXr's suggestion. Due to the time limits, we report the preliminary results for Task 1 and Task 4 below. Results of remaining tasks will be updated later.
> | **Task 1** | PSWE (FS) | FSPooling (FS) | FSW (FS) | SRAL (FS) | PSWE (LJ) | FSPooling (LJ) | FSW (LJ) | SRAL (LJ) |
> | :--- | :--- | :--- | :--- | :--- | :--- | :--- | :--- | :--- |
> | **R@20** | 83.05 | 79.90 | 83.58 | **91.57** | 83.52 | 85.36 | 84.19 | **87.56** |
> | **N@20** | 84.26 | 81.96 | 84.39 | **92.22** | 84.61 | 87.17 | 85.04 | **89.31** |
> | **R@100** | 88.59 | 87.76 | 88.52 | **94.53** | 89.48 | 93.07 | 89.95 | **92.93** |
> | **N@100** | 85.77 | 84.41 | 85.81 | **93.01** | 86.67 | 90.29 | 87.23 | **91.25** |
>
> | **Task 4** | PSWE | FSPool | FSW | SRAL |
> | :--- | :--- | :--- | :--- | :--- |
> | **LDA-1k (AUC)** | 58.36 | 75.67 | 64.56 | **80.94** |
> | **LDA-3k (AUC)** | 78.44 | 70.57 | 75.67 | **87.93** |
> | **LDA-5k (AUC)** | 78.81 | 71.16 | 80.94 | **86.20** |
>
>
> From the results, we observe that FSW performs on par with PSWE. However, our method achieves better performance compared to FSW. We attribute this to the fact that FSW is specialized for "Multisets" (allowing duplicate elements in a set), and its advantages are less pronounced when applied to standard sets (composed of unique elements) used in our context. And our unique adversarial perturbation further improves the set embeddings' quality. We have supplemented details in sections of Related Work and Experiments. Thank you.

---

> > ### Author Response · Authors · 2025-11-24
> > **Response to Reviewer 9yXr (Part V)**
> >
> > Dear Reviewer 9yXr:
> >
> > We have now updated the supplemented experiments for Task 3 as follows. The results indicate that FSW, utilizing ISAB as its backbone, achieves significant improvements and serves as the new second-best model. However, it still underperforms our SRAL method, primarily due to the effectiveness of our unique AEPO mechanism in enhancing embedding quality. We have updated these experimental results and discussions in the revised manuscript. Thank you.
> >
> > | **Task 3** | PSWE | FSPool | FSW | SRAL |
> > | :--- | :--- | :--- | :--- | :--- |
> > | **MLP** | 86.41 | 85.76 | 86.38 | **86.53** |
> > | **ISAB** | 86.85 | 86.88 | 86.93 | **87.31** |

---

> > > ### Comment · Reviewer_9yXr · 2025-11-25
> > >
> > > Thank you for your response. I maintain my favorable rating of the paper.
> > >
> > > With regard to perturbing the projection parameters $\Theta$, what if the perturbation respects the unit-hypersphere requirement of SW? The perturbation can be applied in such a way that $\Theta$ remains on the unit hypersphere. Would that work with your approach?

---

> > > > ### Author Response · Authors · 2025-11-26
> > > > **Follow-up Disucssion with Reviewer 9yXr**
> > > >
> > > > Dear Reviewer 9yXr,
> > > >
> > > > We appreciate your insightful follow-up discussions. As you correctly suggested, it is indeed feasible to apply perturbations to the projection parameters $w$, i.e., $w' = w + \epsilon$, followed by a normalization step $\hat{w} = \frac{w'}{||w'||_2}$. This ensures the perturbation respects the unit-hypersphere constraint required by the SW metric.
> > > >
> > > >
> > > > To verify this, we conducted a preliminary experiment on Task 4 (Topic Set Expansion) and obtained the following results:
> > > >
> > > >
> > > > |Accuracy|SRAL (origin)|SRAL (new)|
> > > > |---|---|---|
> > > > |LDA-1K|80.94|81.06|
> > > > |LDA-3K|87.93|87.89|
> > > > |LDA-5K|86.20|86.15|
> > > >
> > > > with model trainig time cost comparison as follows:
> > > >
> > > > |Time|SRAL (origin) - In total|SRAL (new) - In total|SRAL (origin) - Per-epoch|SRAL (new) - Per-epch|
> > > > |---|---|---|---|---|
> > > > |LDA-1K|14.3s|16.7s|0.5s|0.5s|
> > > > |LDA-3K|23.5s|28.4s|0.5s|0.5s|
> > > > |LDA-5K|31.6s|37.5s|0.6s|0.6s|
> > > >
> > > > We have the following threefold discussions:
> > > >
> > > > - The results indicate that the new variant (perturbing $\Theta$) performs comparably to our original method (perturbing input features), with a slight improvement on LDA-1K. Crucially, both versions consistently outperform all baselines reported in the paper. This confirms that introducing adversarial perturbations (*whether on the data features or the projection process*) can enhance representation quality.
> > > > - From the second table, we observed that while the *per-epoch* computational cost is basically the same, the *total* training time for the new variant is slightly longer, indicating slower convergence. We preliminarily attribute this to the optimization aspect: our original method applies perturbations directly to the embeddings/features, providing a *relatively* direct gradient path for the encoder to optimize against. In contrast, perturbing the projection function ($\Theta$) affects the loss more indirectly, which likely results in the observed lag in convergence.
> > > > - After the team discussions, we find this "projection perturbation" idea highly inspiring. We recognize a potential connection between this approach and Sharpness-Aware Minimization (SAM: Sharpness-Aware Minimization for Efficiently Improving Generalization[ICLR'21]). We plan to explore this connection further in future work.
> > > >
> > > > *Lastly, we sincerely thank Reviewer 9yXr for your early support and favorable rating of our work. Given the competitive nature of ICLR, if you feel our revised version meets your expectations for an improved quality, we would be much grateful if you might consider raising the score. Regardless, we fully respect your final decision and appreciate your dedication to maintaining the high standards of the ICLR community!*
> > > >
> > > >
> > > > Best Regards,
> > > >
> > > > Authors of Submission 2217

---

### Official Review · Reviewer_du3f · 2025-11-01

**Soundness:** 3
**Presentation:** 3
**Contribution:** 3
**Rating:** 6
**Confidence:** 2

**Summary:**

This paper proposes SRAL (Set Representation Auxiliary Learning), a novel framework for set representation learning that explicitly captures inter-set distributional relationships rather than focusing solely on preserving intra-set structures. SRAL introduces a learnable reference distribution O and encodes each set according to its 2-Sliced-Wasserstein distance to this common reference, thereby aligning all sets within a shared geometric space. Moreover, SRAL incorporates a self-supervised objective with adversarial perturbations applied in the aligned feature space to obtain more robust and discriminative set embeddings.

**Strengths:**

- Proposes a theoretically grounded approach to set representation learning based on the Wasserstein distance.
- Achieves superior performance compared to existing methods across various benchmark datasets.

**Weaknesses:**

Motivation for introducing reference set O is a bit unclear to me. In the manuscript, it is only briefly explained: “To learn stable and discriminative set representations, our encoder leverages the distributional distance between an input set and a learnable reference distribution O.” For instance, if my understanding is correct, one could compute an L_wd-like loss without introducing a reference distribution, by directly calculating the sliced Wasserstein distance between each pair of P_i and P_j (with perturbations), and replacing the term -||v_i′ - v_j″||_2 with that sliced Wasserstein distance. Could the authors clarify this design choice or point out what I might be missing?

**Questions:**

- How were the hyperparameters chosen, and how computationally costly is the proposed method? The approach requires tuning the hyperparameters λ1 and λ2, and according to Table E.1, the optimal values of λ1 vary considerably across different tasks (from 5e-1 to 1e-3). For comparison, do the baseline methods such as FSPool and CrossCBR in the benchmark tables also require similar levels of hyperparameter tuning to achieve the reported performance? It would be interesting to see how the proposed method performs in terms of computational efficiency and cost.

- In Eq. (5), the mapping g+ appears to be non-differentiable. Did you use any soft approximation of the sorting operation, or how do you propagate the gradients with respect to the parameters.

- Typos and minor notes
  - L.107: g: R^d -> R^d
  - L.155: should be "true data distribution P_i" (without a hat)
  - L.159: (perhaps clear from context, but to confirm) inside delta, it should be theta(x) >= theta(z), or x and z should be one-dimensional scalars.
  - L.257: the correct reference should be https://arxiv.org/abs/1412.6572


I will adjust the score based on replies from the authors.

---

> ### Author Response · Authors · 2025-11-21
> **Response to Reviewer du3f (Part I)**
>
> We appreciate Reviewer du3f for your insightful feedback. Below, we provide point-by-point responses to address the raised concerns, with all corresponding revisions below highlighted in blue in the manuscript. We hope these clarifications and modifications satisfactorily address the concerns.
>
>
>
> **W1**:
> - Thanks Reviewer du3f for pointing this out. Yes, technically, it is feasible to directly model the problem in the manner you suggested, and we indeed explored this approach in our early model designs. However, we subsequently observed that the model exhibited highly *unstable training convergence, with significant performance fluctuations*.
> - Upon careful analysis, we identified that the reason is frm our set encoding mechanism (Eq. (6)), which requires features from two sets as input. Without anchoring one of these inputs as a fixed reference, for different set pairs, e.g., ($P_1$, $P_2$) and ($P_1$, $P_3$), the model generates two representations for the same set $P_1$, leadning to representational inconsistency. This problem is further exacerbated by the necessity of data shuffling during training to prevent overfitting. Since we cannot maintain an ordered set pair sequence, e.g., ($P_1$, $P_2$), ($P_1$, $P_3$), ($P_1$, $P_4$)..., the same set is repeatedly encoded differently with different input sets in this unpredictable sequence.
> - After thorough consideration, we then chose to introduce a fixed reference, even at the cost of potentially sacrificing optimal performance on certain tasks. It is worth noting that the embedding of the reference itself remains learnable. However, it is held relatively stable within each batch and only updated via gradient descent after the batch computation is complete. This design strikes a balance: the reference provides the necessary stability for consistent set encoding within a batch, while still allowing the model to adapt its representation through learning. Therefore, this approach yields more stable training convergence and empirical performance.

---

> ### Author Response · Authors · 2025-11-21
> **Response to Reviewer du3f (Part II)**
>
> **Q1**:
> Thank you Reviewer du3f for the crucial questions.
>
> **(1) Hyperparameter Selection**:
> Our experimental protocol employs a standard train-validation-test split. Hyperparameters for all methods, including our approach and all baselines (FSPool, CrossCBR, etc.), were tuned via grid search on the validation set following the same procedure.
>
> **(2) Computational Efficiency**:
> - Thanks Reviewer du3f for the reminder. We report the results against the top-2 competitively performing baselines. Results for Task 1 are presented below, with complete runtime comparisons across all 4 tasks added to Appendix E.9 in the revised manuscript.
>
>     |  | Friendster (all)| LIVEJ (all)| Friendster (per-epoch)| LIVEJ (per-epoch)|
>     |-----|-----|-----|-----|-----|
>     |PSWE|  3.1h  |  5.1h |   1.6min  |  2.7min |
>     |FSPool|  2.9h |  6.5h |   1.5min  |  2.1min |
>     |SRAL|  3.9h  |  7.7h |   5.6min  |  7.9min |
>
>     While our model incurs higher per-epoch computational cost, the proposed Adversarial Encoding Perturbation (AEPO) mechanism substantially promotes convergence. Under an early-stopping strategy, the total training time actually remains comparable to baseline methods. A detailed convergence analysis substantiating this claim is provided in Section 4.3.1.
>
> - To further elucidate the computational breakdown, we report the runtime contribution of each module on a per-epoch basis over Friendster dataset:
>
>     | | SRAL (complete) | SRAL (SFE module)  | SRAL (AEPO) | SRAL (Main loss) |
>     | :--- | :---: | :---: | :---: | :---: |
>     | **Friendster** | 5.6min | 1.8min |  3.3min | 0.1min |
>
>     As expected, the AEPO module costs more computation time. However, considering its effectiveness in promoting faster convergence and enhanced training stability, we believe such cost is an acceptable trade-off.
>
> - In addition, we also conducted scalability experiments to assess the performance and runtime costs across varying data sizes. As detailed in Appendix E.8, the results demonstrate that our approach maintains reasonable computational scaling properties with respect to dataset size.
>
> We hope the above analyses adequately address the reviewer's concerns.
>
> **Q2**: Thanks Reviewer du3f for your insightful question. The rank-based mapping operation in Eq.(5) is originally non-differentiable. Our implementation relies on the property of Pytorch's automatic differentiation framework that essentially functions as an implicit Straight-Through Estimator: (1) For the input set $V_i$, the gradient path is differentiable. This is because our implementation utilizes the sorted **`element values`**, not just their discrete ranks. The operation of sorting by value is differentiable in Pytorch's autodiff frameworks (the gradient is passed "straight-through" to the corresponding pre-sorted elements). This ensures that the element embeddings in $V_i$ can be trained by both the main and auxiliary losses. (2) For the reference set $V_O$, in backpropagation, the gradient path through the **`rank`** function is discrete, so its gradient is cutted. However, in our implementation, we use $V_O$ as a learnable point in the distance computation, which provides a parallel and fully differentiable path. Then during backpropagation, Pytorch's autodiff mechanism just ignores the cutted "rank path" and routes the gradient along this "differentiable path" to $V_O$. This allows the parameters to be trained effectively as well.
>
> **Q3**: Thanks Reviewer du3f for your detailed guidance.
> (1) We wish to clarify that this operation refers to the projection of a high-dimensional distribution into a one-dimensional space. This step is essential because the one-dimensional setting is a *prerequisite* for computing the later closed-form solution $g^+$. We have added a supplementary explanation in the revision (L.116) for better clarity:
> > L.125: "For the projection function $g: \mathbb{R}^d \rightarrow \mathbb{R}$, $P^{g}$ represents the push-forward of ${P}$ with $g$ in a one-dimensional space, ..."
>
> (2) We have corrected this as suggested.
>
> (3) We thank the reviewer for pointing out this issue. We agree that the $\ge$ symbol is ambiguous when applied to $d$-dimensional vectors $\boldsymbol{x}$ and $\boldsymbol{z}_{i,k}$. Our intention was to define a **`Multivariate CDF`** in $\mathbb{R}^d$ space, where the comparison is performed element-wise. We have clarified this in the revision as follows:
> > L.178: "Therefore, we can define the empirical Multivariate Cumulative Distribution Function (CDF) of $P_i$ as: $F_{P_i}(\mathbf{x}) = \frac{1}{|S_i|} \sum_{k=1}^{|S_i|} \delta(\mathbf{x} \geq \mathbf{z}_{i, k})$, where $\delta(\cdot)$ is the indicator function$^1$."
> >
> and
>
> > "$^1$The comparison $\mathbf{x} \geq \mathbf{z}\_{i, k}$ is performed element-wise. That is, $\delta(\cdot)$ returns 1 if and only if $\mathbf{x}\_{j} \geq \mathbf{z}\_{i, k, j}$ for all dimensions $j=1, \ldots, d$, and 0 otherwise."
>
> (4) We have corrected the reference. Thank you.

---

> ### Author Response · Authors · 2025-11-25
> **Follow-up on Rebuttal**
>
> Dear Reviewer du3f,
>
> We sincerely appreciate the time and effort you dedicated to reviewing our work.
>
>
> Mindful of the potential Thanksgiving break and the rebuttal concluding in about one week, we wanted to kindly reach out to ensure our response fully addresses your concerns. If there are any remaining questions or points requiring further clarification, we would be very pleased to provide further explanations in the following time.
>
> We have now revised the manuscript to carefully address all the concerns. Should you find that these clarifications meet your expectations, we would be most appreciative if you could consider updating your final evaluation accordingly.
>
> Thank you once again for your time, expertise, and support.
>
> With sincere appreciation,
>
> Authors of Submission 2217

---

> ### Author Response · Authors · 2025-11-27
> **A Further Update about Q3 (1), (3)**
>
> Dear Reviewer du3f,
>
> After discussion with Reviewer jq5C, we have two further updates (please refer to our latest revision) as follows:
>
> - (1): We acknowledge that this confusion arose from overloaded notation. In Eq. (1) and the subsequent theoretical context, $g$ and $g^+$ denote the transport plan and the optimal transport map between distributions $P$ and $Q$, which operate within the same dimensionality. However, in L.130, the symbol $g$ was inadvertently used to denote a projection function, creating ambiguity. To resolve this, we have renamed the function in line 126 to $\theta$. This correction ensures consistency with the projection notation used throughout the rest of the paper (e.g., as defined in Eq. (2) and Section 3.2).
>     > L. 130: For the projection function $\theta: \mathbb{R}^d \rightarrow \mathbb{R}$, $P^{\theta}$ represents the push-forward of ${P}$ with $\theta$ in a one-dimensional space, defined as $P^{\theta}({Y}) = P({{x}: \theta({x}) \in {Y}}) = P(\theta^{-1}({Y}))$.
>
> - (3): We have now concluded that the Multivariate CDF is not strictly necessary there (*Our original intention was to present the general form first, allowing the one-dimensional CDF in the sliced setting as a special case*). Therefore, we have omitted the explicit definition for this Multivariate CDF, and instead, we now directly introduce the CDF for the projected 1D distribution directly, i.e., in the context of Eq. (4) of our latest revision, as follows:
>     > L. 202: "Here the CDF of $O^\theta$ is: $F_{{O}^\theta}({x}) = \frac{1}{H} {\sum}_{h=1}^{H} \delta(x \geq \boldsymbol{w}^\mathsf{T}\boldsymbol{z}_h)$, where $\delta$ returns 1 for zero input and 0 otherwise."
>
> We thank Reviewer du3f for your attention to these updates, as well as the time and support for our work.
>
> Best Regards,
> Authors of submission 2217

---

### Author Response · Authors · 2025-12-01
**Summary of Rebuttal Updates and Reviewer Consensus**

Dear Area chairs,

Due to the technical rollback on OpenReview, the final discussion time window was lost. Below is a summary of the consensus reached so far with active reviewers and the comprehensive actions taken to address all concerns.

**Reviewer 9yXr: 6 (before rebuttal) $\rightarrow$ "Maintained favorable rating" (after rebuttal)**

- **Concerns**: distinction between data-level and feature-level perturbations; comparison with recent baselines (FSW, DIEM); clarify SFE module vs. PSWE; feasibility of perturbing projection parameters; notation clarity.
- **Our Rebuttal**: clarified that AEPO applies learned adversarial perturbations at the feature level to maximize robustness; conducted new experiments showing SRAL outperforms FSW and DIEM on Tasks 1 & 4; explained SFE uses concatenation to better approximate Wasserstein distance; explored projection-parameter perturbation as a promising future direction.
- **Paper Revision Updates**: have revised Abstract and Introduction to clarify the perturbation mechanism (Sec. 1); updated experimental tables with FSW/DIEM results; added discussion on projection perturbations (Appendix). -
- **Reviewer’s Stance**: Reviewer 9yXr acknowledged the clarifications and new results, explicitly stating: "I maintain my favorable rating of the paper".


---

**Reviewer jqC5: 6 (before rebuttal) $\rightarrow$ "Maintained positive score and hope for acceptance" (after rebuttal)**
- **Concerns**: missing comparisons to OT-based methods (RepSet, ApproxRepSet); definition of Multivariate CDF; clarify learning of reference set $\mathcal{O}$; justification of "provably aligned" claim; request for illustrative examples.
- **Rebuttal**: have added extensive comparisons with RepSet and ApproxRepSet confirming SRAL's superiority; simplified formulation explanation by removing Multivariate CDF in favor of 1D CDF; clarified $\mathcal{O}$ comprises trainable embeddings; refined claim to "positively correlated" with SW metric; added concrete examples (e.g., product bundles).
- **Paper Revision Updates**: expanded Related Work (Sec. 2); rewrote Preliminaries with new "Illustrative Example" (Sec. 3.1); corrected transport notation and simplified Propositions (Appendix D); refined "alignment" claims (Sec 3.3)7.
- **Reviewer’s stance**: Reviewer jqC5 engaged deeply in discussion and updated **`the original review with a "post-rebuttal comment"`** stating "maintain a positive score and hope for the paper's acceptance".


---

**Reviewer du3f: 6 (before rebuttal) $\rightarrow$ No follow-up response yet (due to rollback)**

- **Concerns**: motivation for using a reference set $\mathcal{O}$; computational cost and efficiency; differentiability of the sorting operation; hyperparameter sensitivity.
- **Rebuttal**: explained $\mathcal{O}$ anchors training stability compared to unstable direct set-pair comparisons; provided runtime analysis showing comparable total training time due to faster convergence; clarified use of PyTorch’s implicit Straight-Through Estimator for gradient passing; provided hyperparameter grid search details.
- **Paper Revision Updates**: added detailed runtime and convergence analysis (Sec. 4.3.1 & Appendix E.9); clarified differentiability of rank-based mapping (Sec. 3.2).
- **Reviewer’s stance**: no follow-up comment was posted before the system rollback; however, all technical soundness questions and concerns were fully answered in our response.

---

**Reviewer kdXr: 4 (before rebuttal) $\rightarrow$ No follow-up response (due to rollback)**
- **Concerns**: writing clarity and heavy notation; lack of baselines (PoT, Set2Box); marginal gains on some tasks; integration of LLMs; figure readability.
-- **Rebuttal**: substantially simplified notation and presentation; added requested comparisons against PoT and Set2Box; provided standalone results for Task 2 to isolate SRAL contribution; discussed synergy with LLMs as feature extractors/adapters.
- **Paper Revision Updates**: rewrote Sections 2 & 3 for clarity; enlarged Figures/Tables; added comparative results to Experiment Section; added LLM discussion to Conclusion.
- **Reviewer’s stance**: no follow-up comment was posted before the system rollback; All concerns were addressed in revision.

---
**Overall Summary**
Due to unexpected OpenReview system rollback, the follow-up confirmations from Reviewers du3f and kdXr were unfortunately interrupted. However, Reviewers 9yXr and jqC5 explicitly commented to maintain their positive scores for acceptance after going through our rebuttals. Crucially, all concerns across all four reviews have been adequately addressed, and all clarifications, additional experiments, and textual improvements requested have been fully incorporated into the paper revision.

Lastly, we extend our sincere gratitude to all reviewers and both Area Chairs for the time, dedication, constructive feedback to our work and all the support in navigating the technical challenges of this review cycle.

---

### Meta-Review · Area_Chair_tVC6 · 2026-01-02

**Summary:**

The Reviewers have raised concerns regarding
(1) motivation for introducing a reference set O;
(2) how and where adversarial perturbations are applied;
(3) the overlap between the SFE module, a key component in the proposed learning framework, and the prior art PSWE;
(4) the novelty of applying optimal transportation in set representation learning;
(5) presentation and clarity of the proposed methods;
(6) limited comparison with baseline benchmarks; and
(7) marginal gains for 2 out of 4 tasks.

In addition, there are a few technical questions.

**Reviewer Concerns:**

After going through all discussions and responses and reading part of the revised manuscript, I do believe that the above concerns have been adequately addressed by the rebuttal.

**Reviewer Scores:**

The three reviewers with their original rating of 6 would have likely maintained their rating of 6. The reviewer with the original rating of 4 would have likely increased their score to 6.

---

### Decision · Program_Chairs · 2026-01-26

Accept (Poster)